# Artificial neural networks for short-term forecasting of cases, deaths, and hospital beds occupancy in the COVID-19 pandemic at the Brazilian Amazon

**Marcus de Barros Braga**[1]*, **Rafael da Silva Fernandes**[2], **Gilberto Nerino de Souza, Jr**[1], **Jonas Elias Castro da Rocha**[1], **Cícero Jorge Fonseca Dolácio**[3], **Ivaldo da Silva Tavares, Jr**[4], **Raphael Rodrigues Pinheiro**[5], **Fernando Napoleão Noronha**[2], **Luana Lorena Silva Rodrigues**[6], **Rommel Thiago Jucá Ramos**[7], **Adriana Ribeiro Carneiro**[7], **Silvana Rossy de Brito**[8], **Hugo Alex Carneiro Diniz**[9], **Marcel do Nascimento Botelho**[10], **Antonio Carlos Rosário Vallinoto**[7]

**1** Paragominas Campus, Universidade Federal Rural da Amazônia, Paragominas, Pará, Brazil,
**2** Parauapebas Campus, Universidade Federal Rural da Amazônia, Parauapebas, Pará, Brazil, **3** Forest Engineering and Technology Department, Universidade Federal do Paraná, Curitiba, Paraná, Brazil,
**4** Forestry Engineering Department, Universidade Federal de Viçosa, Viçosa, Minas Gerais, Brazil, **5** Belém Campus, Universidade Federal Rural da Amazônia, Belém, Pará, Brazil, **6** Postgraduate Program in Health Sciences, Institute of Collective Health, Universidade Federal do Oeste do Pará, Santarém, Pará, Brazil, **7** Institute of Biological Science, Universidade Federal do Pará, Belém, Pará, Brazil, **8** Cyberspace Institute, Universidade Federal Rural da Amazônia, Belém, Pará, Brazil, **9** Institute of Educational Sciences, Universidade Federal do Oeste do Pará, Santarém, Pará, Brazil, **10** Socio-Environmental Institute of Water Resources, Universidade Federal Rural da Amazônia, Belém, Pará, Brazil

* marcus.braga@ufra.edu.br

**Data Availability Statement:** All relevant data are within the paper and its Supporting information files.

## Abstract

The first case of the novel coronavirus in Brazil was notified on February 26, 2020. After 21 days, the first case was reported in the second largest State of the Brazilian Amazon. The State of Pará presented difficulties in combating the pandemic, ranging from underreporting and a low number of tests to a large territorial distance between cities with installed hospital capacity. Due to these factors, mathematical data-driven short-term forecasting models can be a promising initiative to assist government officials in more agile and reliable actions. This study presents an approach based on artificial neural networks for the daily and cumulative forecasts of cases and deaths caused by COVID-19, and the forecast of demand for hospital beds. Six scenarios with different periods were used to identify the quality of the generated forecasting and the period in which they start to deteriorate. Results indicated that the computational model adapted capably to the training period and was able to make consistent short-term forecasts, especially for the cumulative variables and for demand hospital beds.

## Introduction

The World Health Organization, on January 30, 2020, declared the Severe Acute Respiratory Syndrome—CoronaVirus 2 (SARS-CoV-2) pandemic. The first cases were reported in China,

**Funding:** The author(s) received no specific funding for this work.

**Competing interests:** The authors have declared that no competing interests exist.

in the city of Wuhan, in December 2019. Up to June 30, 2020, the world had 10,176,328 confirmed cases and 500,812 deaths [1]. SARS-CoV-2 is the etiological agent of the disease known as COVID-19 (Corona Virus Disease 19), whose main symptoms are fever, cough, myalgia or fatigue, sputum, and dyspnea [2].

At that time, Brazil had 1,402,041 confirmed cases and 59,594 deaths [3]. The first reported case occurred in the city of São Paulo on February 26, 2020, and, 23 days later, authorities recognized the occurrence of community transmission in the national territory [4].

In Pará, a state in the Brazilian Amazon region, the first COVID-19 case was notified in the capital, Belém, on March 18, 2020. Since then, state authorities have implemented measures of physical distance, isolation, quarantine, and lockdown, in addition to the distribution of complementary drugs, implantation of clinics and field hospitals [5]. In this context, several studies have contributed significantly to understand the biological, physiological, and climatic factors that can influence the spread of the virus [2,6–14].

Some studies also applied mathematical models to provide the quantitative framework in which scientists can assess hypotheses about potential underlying mechanisms. This model explains patterns in data observed at different spatial and temporal scales [15]. These models are useful for assessing the impact of interventions, optimizing the control strategies' impact, and generating short and long-term forecasts.

The growing use of mathematical models for epidemic forecasts has proven the importance of obtaining reliable models to capture the basic characteristics of pathogens transmission in specific social contexts. Some studies have already shown that artificial intelligence techniques can be promising and support the fight against the COVID-19 pandemic progression [16,17].

That said, artificial neural network (ANN) is a technique that can be used to model epidemiological phenomena, forecast epidemic peaks, and estimate the dimension of the risk and scope of diseases [18–21]. The main characteristic of ANN is self-learning without prior knowledge of the complex non-linear relationships that exist between the input and output variables [22]. This is due to the massive and parallel processing of neurons and the tolerance to noise [23]. In addition, this technique captures small distortions in the observed data and transfers them for projections differently from mechanistic models [24]. Another advantage is that this type of approach also makes it possible to use several predictor variables simultaneously, such as demographic data and incidence curves, which helps in capturing the dynamics of virus transmission in the cities over time [25,26].

ANN has shown good forecasting results in emerging epidemiological outbreaks such as Ebola, Zika, and Middle East Respiratory Syndrome [27,28]. Additionally, Yang et al. focused on the COVID-19 outbreak in China between January and March 2020, using models based on ANN and obtaining remarkable results when compared to the SEIR compartmental model (Susceptible Exposed Infectious Recovered) [29]. Tamang, Singh & Datta, showed that ANN is an efficient technique to process large data sets when modeling the number of COVID-19 cases from India, the USA, France, and the United Kingdom [30].

ANN was also used in short-term forecasting to predict the predominance of the COVID-19 epidemic in Egypt [31]. This study indicated good results when compared to the statistical model ARIMA (Autoregressive Integrated Moving Average), suggesting good agreement with the historical data in up to 17 days of forecasting of confirmed cases.

In this pandemic context, the need to anticipate substantial increases in the capacity of standard hospital beds and Intensive Care Unit (ICU) beds is also relevant to prepare workflows in advance for the patients' diagnosis and fast isolation [32,33]. Hospital bed estimation proposals have been put forward to identify the demand for ICU beds in the USA and China [34]. However, no studies have been found in the literature that modeled the demand for hospital beds in short-term forecasting, making this study a pioneer in the area.

This study sought to contribute to a broader understanding of COVID-19 transmission dynamics, evaluating its progression over time in the State of Pará. This strategy, addressed in Fernandes, aims to minimize uncertainties associated with forecasts [35]. For this reason, ANNs were trained with data from 6 different moments, incorporating the ability to assess the quality of forecasting at different epidemic stages in their structure. The assessment of these different scenarios can help public health authorities to implement more effective interventions.

To this end, the present study used ANN to forecast the number of confirmed cases and cumulative deaths, the number of confirmed cases and daily deaths, as well as the standard hospital beds and ICU beds occupancy during the COVID-19 pandemic in Pará State. This research specifically answers two main questions: i) is there an improvement in the quality of the forecasting when inserting new data? ii) in how many days does the generated forecasting by the neural network begins to deteriorate?

## Materials and methods

### Study area

Pará is the Brazilian State with the lowest municipal human development index. It is in 24th place (0.646) among the 26 states of Brazil and the Federal District [36]. Pará (Fig 1) has 1,248,000 km$^2$ and more than 8 million inhabitants, where 9.1% are over 60 years old [37].

According to Köppen's climate classification, Pará is in a tropical zone and has regions with tropical rainy (Af), tropical monsoon (Am), and dry winter (Aw) climates [38]. Still, according to these authors, the state annual mean air temperature can vary from 24˚C (75,2˚F) to more than 26˚C (78,8˚F), and the total annual rainfall between 1600 millimeters and 3100 millimeters.

It is possible to realize that there was a high underreporting of COVID-19 in the northern region of Brazil during the period evaluated [39]. This refers to the premise that the probable factors of these underreporting are mainly due to the low number of tests—increased by the number of asymptomatic individuals who do not seek the health system for testing.

These factors may have an impact on an appropriate testing performance and delay the spread of data related to COVID-19 confirmed cases, especially in the poorest regions of the country, such as the northern region, although the underreporting of deaths is relatively smaller [40].

### Data collection

For this study, our data selection encompassed a particular period. It starts on March 18, 2020, the date on which Brazil had the first COVID-19 case notified, and it finishes on June 30, 2020 (Fig 2). The data on confirmed cases and deaths by COVID-19 used were collected in the official database of the Pará's government [41].

Until June 30, 2020, the Pará State accounted for 105,855 cases and 4,960 deaths [42]. These data showed that the epidemic curve of the number of cumulative deaths was in a third growth phase, characterized by a decrease in the growth rate and a stabilization trend. The cumulative case curve, after the initial exponential growth phase, showed a linear growth trend.

The peak of daily deaths in Pará occurred 49 days after the first death (May 05, 2020). After this date, a deceleration in the death curve for 56 days was observed. The daily case curve peak was in the first half of May, but it was still early to assume a deceleration in the number of infected cases for the same period.

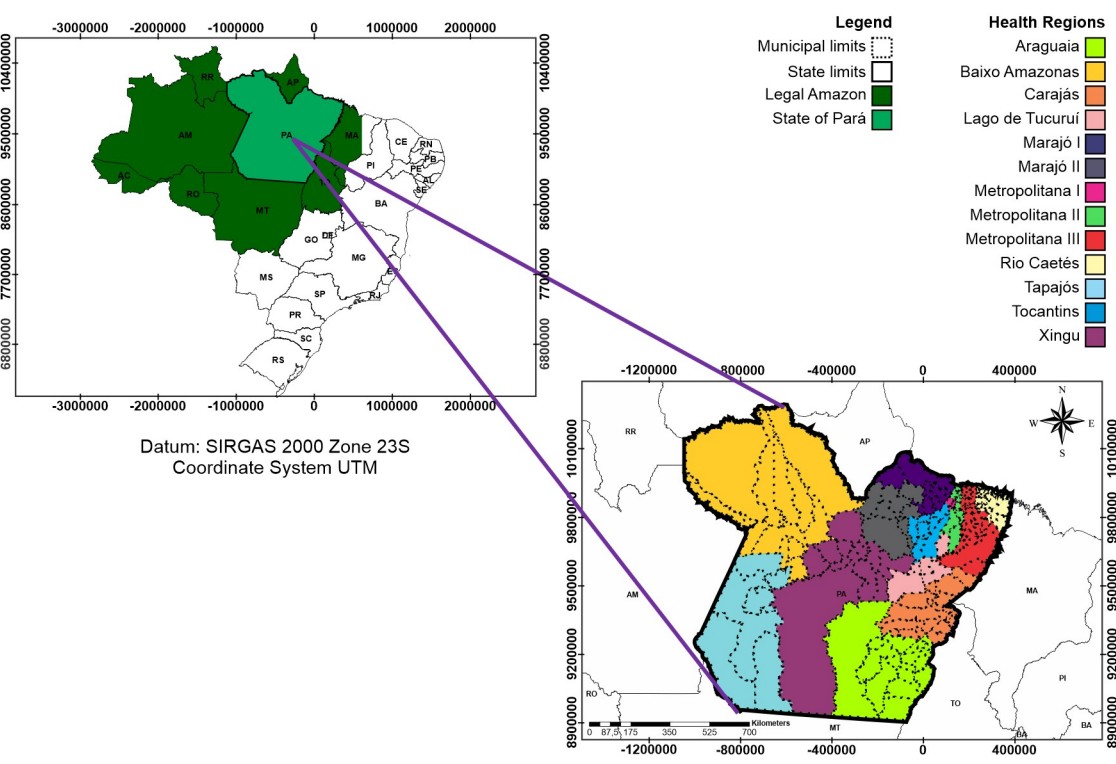

**Fig 1. Location map of the Pará State, Brazil.**

Regarding hospital beds, the occupancy peaked in mid-May when it began presenting a slight downward trend. As for ICU beds occupancy rates, dedicated to severe COVID-19 cases, there was a growth until early June and subsequent stabilization.

## ANN modeling

Different supervised ANNs architectures were trained to model the following output variables together (Fig 3): the number of cases and cumulative deaths (cumulative variables); the number of cases and daily deaths (daily variables); and the standard hospital beds and ICU beds occupancy, in six different scenarios.

The dates and number of days used for training the ANNs varied among the scenarios. However, the initial date was March 18th for all scenarios. The final dates and the number of days of the training periods for scenarios 1 to 6, were respectively: May 12th (56 days), May 19th (63 days), May 26th (70 days), June 6th (77 days), June 9th (84 days), and June 16th (91 days).

The trained ANNs were the multilayer perceptron type, with feedforward architecture and a hidden layer, using the Broyden-Fletcher-Goldfarb-Shanno (BFGS) iterative algorithm that can be used for solving optimization problems, from the Statistica 13.5 software [43]. The processing took place on a computer with two Intel® Xeon® Silver 4114 Processor (13.75 M Cache, 2.20 GHz), 20 GB of RAM, 10 cores, and 64-bit Windows 10 Pro Operating System.

The BFGS memoryless quasi-Newton was successfully used for minimizing errors on artificial neural networks. The Quasi-Newton method is a method that is used when the calculation of the Hessian matrix is difficult or time-consuming. This method has a rapid convergence when compared with the method of gradient descent [44].

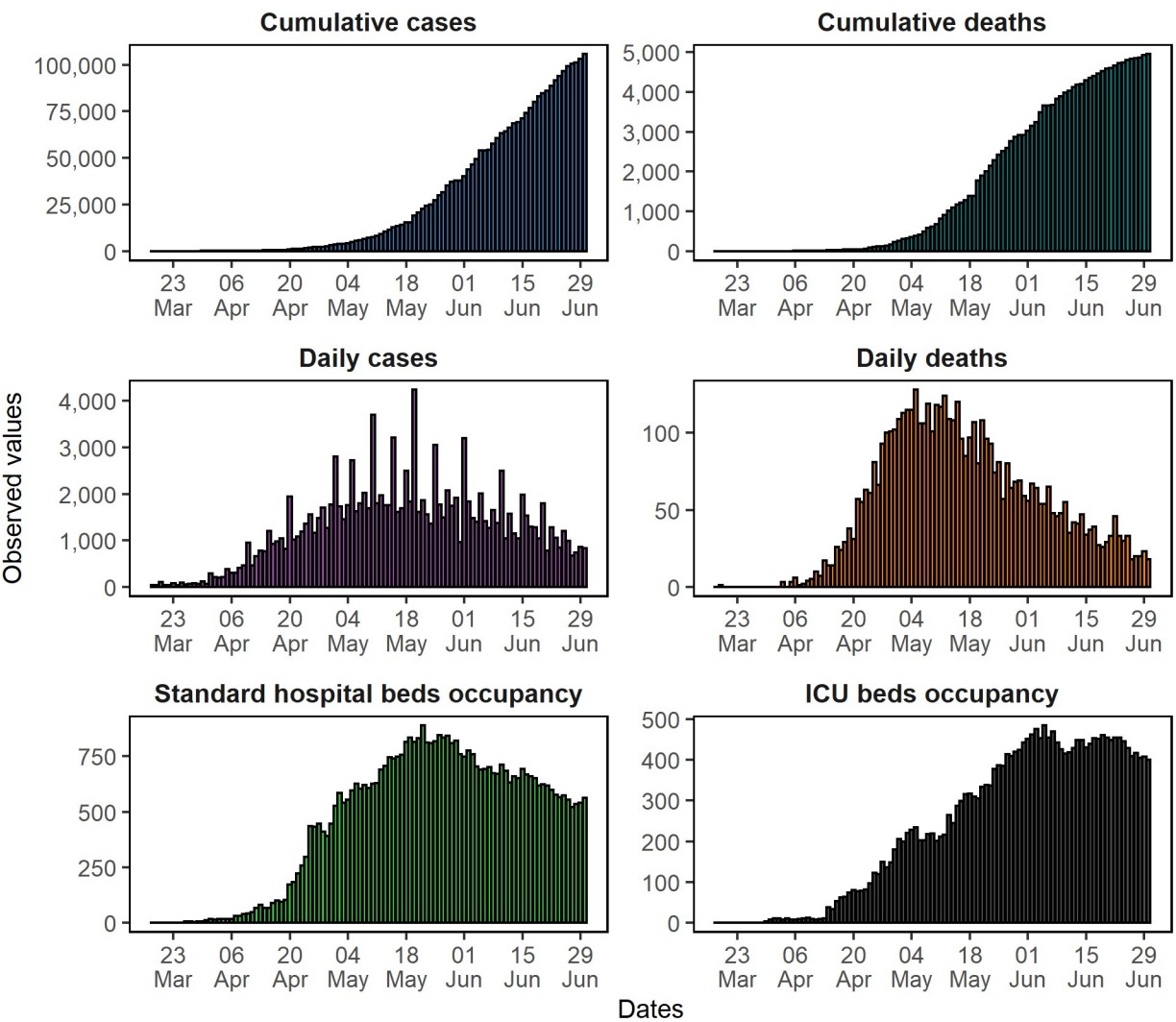

**Fig 2. Cases, deaths, and hospital beds occupancy caused by COVID-19 in the Pará State, Brazil.**

One hundred and fifty-nine neurons were used in the input layer of the ANNs trained to forecast the cumulative variables (S1 Appendix). Two of them corresponded to the quantitative variables' standardized values: municipal demography and occurrence date. The other 157 neurons corresponded to the categorical variables: name of State's municipalities (144 municipalities) and the names of the health regions (13 regions).

In the ANN architectures trained to predict daily variables, 15 neurons were used on the input layer (S2 Appendix). Two of them corresponded to the quantitative variables' standardized values: region health demography and occurrence date. The 13 other neurons corresponded to the categorical variables: the names of the health regions. To predict the standard hospital beds and ICU beds occupancy, two neurons corresponding to the standardized values of the daily death numbers predicted by the best ANN and occurrence date were used (S3 Appendix).

To model the cumulative and daily variables of cases, deaths, and hospital beds, an adaptation to the traditional Fletcher-Gloss method [45] was performed to define the tested neurons

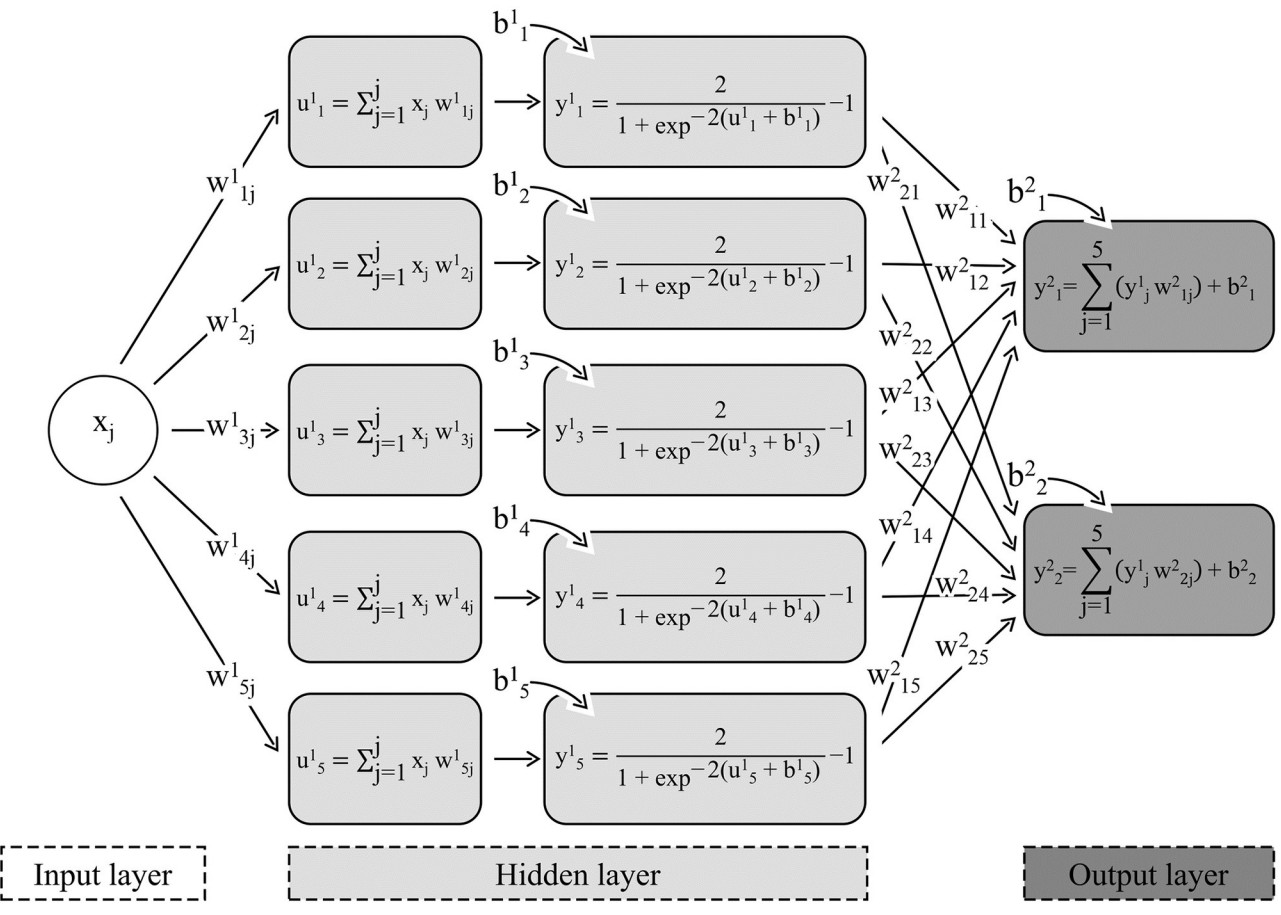

**Fig 3. ANN architecture with 1 neuron at input layer, 5 neurons and hyperbolic tangent activation function at hidden layer, and 2 neurons and linear activation function at output layer.** $x_j$ is the standardized or binary scale output of the j-th neuron of input layer when quantitative or categorical variables were used, respectively. $w^1_{ij}$ is the synaptic weight that connects the output of the j-th neuron of the input layer to the input of the i-th neuron of the hidden layer. $u^1_i$ is the result of the scalar product between $x_j$ and $w1ij$. $b1i$ is the bias added to the i-th neuron of the hidden layer. $y^1_i$ is the output of the i-th neuron from the hidden layer. $w^2_{ij}$ is the synaptic weight that connects the output of the j-th neuron of the hidden layer to the input of the i-th neuron of the output layer. $b^2_i$ is the bias added to the i-th neuron of the output layer. $y^2_i$ is the output of the i-th neuron from the output layer.

range in the hidden layer of the ANNs (Eq 1).

$$(2 \ \cdot \ \sqrt{n} + n_2) - \alpha \ \leq n_1 \leq \alpha + (2 \ \cdot \ \sqrt{n} + n_2) \tag{1}$$

Being,

$$\alpha = \begin{cases} 5, \ \text{for cases and deaths} \\ 4, \ \text{for hospital resources} \end{cases}$$

Where n is the number of neurons in the input layer; $n_1$ is the number of tested neurons in the hidden layer; and $n_2$ is the number of neurons in the output layer.

This adaptation was performed because the traditional way would result in a long processing time to model the cumulative and daily variables. It would occur due to the number of different architectures that would be trained according to the number of input variables. In contrast, if the traditional method was used for all hospital beds few architectures would be tested and when using $\alpha = 5$, the minimum would be equal to 0.

Four activation functions were tested on the hidden and output layers: identity, exponential, logistic, and hyperbolic tangent. For that reason, considering all the combinations between the number of neurons and activation functions, 176 different ANNs were trained to model the cumulative and daily variables, and 144 ANNs were trained for hospital beds.

## Goodness-of-fit

For each application and data set observed, it is necessary to choose the most appropriate technique among the many available. In this case, different performance metrics can be used as selection criteria. Zeng and Wen used Mean Absolute Deviance (MAD) and Mean Squared Prediction Error (MSPE) [25,46], and, in an epidemiological context, Chowell used the Root Mean Squared Error (RMSE), Mean Squared Error (MAE), and Mean Absolut Percent Error (MAPE) [15].

In this study, the data of each scenario were divided into training set (70%), validation set (15%), using the early stopping method, and test set (15%) to evaluate the qualities of the trained ANNs. Among the ANNs trained to each scenario, the five best were selected based on the lowest average of Sum of Squares Error (SSE; Eq 2) and the highest Pearson's linear correlation coefficient between the real observed values ($r_{y\hat{y}}$; Eq 3), considering the training set.

$$SSE = \sum_{i=1}^{n} (\hat{y}_i - y_i)^2 \tag{2}$$

$$r_{y\hat{y}} = \frac{cov(y_i, \hat{y}_i)}{\sqrt{s^2(y_i).s^2(\hat{y}_i)}} \tag{3}$$

Where $\hat{y}_i$ is the predicted values; $y_i$ is the observed values; cov is the sample covariance; and $s^2$ is the sample variance.

After selecting five ANNs on the training phase, the best ANN of each scenario was selected based on the Weighted Value (WV; Eq 4) of the accuracy measures calculated from the test dataset: $r_{y\hat{y}}$, RMSE (Eq 5), in percentage, MAE (Eq 6), bias (Eq 7), and Normalized Root Mean Squared Error (NRMSE; Eq 7).

$$WV = \sum_{i=1}^{n} nr_i \cdot p_i \tag{4}$$

$$RMSE = 100 \frac{\sqrt{\frac{1}{n}\Sigma_{i=1}^{n}(\hat{y}_i - y_i)^2}}{\bar{y}_i} \tag{5}$$

$$MAE = \frac{\Sigma_{i=1}^{n}|\hat{y}_i - y_i|}{n} \tag{6}$$

$$bias = \frac{\Sigma_{i=1}^{n}(\hat{y}_i - y_i)}{n} \tag{7}$$

$$NRMSE = 100 \frac{\sqrt{\frac{1}{n}\Sigma_{i=1}^{n}(\hat{y}_i - y_i)^2}}{s(y_i)} \tag{8}$$

Where $nr_i$ is the number of records which obtained the ith placement; $p_i$ is the weight of the

ith placement; n is the total number of observed values; $\hat{y}_i$ is the predicted values; $y_i$ is the observed values; $\bar{y}_i$ is the mean of the observed values; and s is the sample standard deviation.

WV ordinated the ANN from a ranking generated with weights assigned based on the efficiency of each accuracy measure [47]. In this case, the weight equal to 1, was assigned to the most accurate ANN, the weight equal to 2 to the second most accurate, and successively in each accuracy measurement. However, when two or more ANNs presented the same value by accuracy measure, the weight assigned corresponded to the average value of the ranking.

Finally, the ANN that obtained the lowest sum of WV was considered the best and, therefore, was used to carry out our forecast. Particularly for cumulative variables that presented decreasing forecasts, the second-best ANN was used.

## Forecasting validation

The forecasts behavior was evaluated in different moments because normally, it is certain that the initial growth phase of an epidemic follows exponential growth dynamics. This situation suggests an overestimated epidemic forecasting trend.

In this context, scenarios 1 to 6 fitted with data of different time periods, provided useful information about the forecast's quality. The number of days of these forecasts to the scenarios 1 to 6 were: 49, 42, 35, 28, 21, and 14 days, respectively.

In this perspective, the forecasts made by the best ANN of each scenario were evaluated by comparing the RMSE, in percentage, and the percent bias (pbias; Eq 9). They were calculated for 7 and 14 days because they were the only coincident intervals in all scenarios. This analysis allowed to infer on the premise of improving the quality of the forecasts with the insertion of data.

$$\text{pbias} = 100 \frac{\sum_{i=1}^{n} \left( \frac{\hat{y}_i - y_i}{y_i} \right)}{n} \tag{9}$$

Where, $\hat{y}_i$ is the forecast values; $y_i$ is the observed values, and n is the total number of observed values.

The diagnosis of the percentual residuals was also performed to evaluate the forecasting quality. This analysis allowed to identify in how many days the forecasting lost its validity and deteriorated. It has also provided us signs on the technique's consistency by making it possible to evaluate the residue distribution.

This study's error acceptance percentage was defined at ± 15% for cumulative variables and hospital beds. For daily variables, where there is a greater dispersion of data, the 0.01 and 0.99 percentile of percentage errors calculated in the last 28 days of ANN training were used to establish the lower and upper limit of forecasting errors. This approach preserved the probability coverage during the forecasting interval, even though it is not symmetric around the punctual forecasting.

The main advantage of using this percentile is the construction of an interval derived from the errors obtained during the training phase. Moreover, the 28 days' period was defined by verifying a random pattern of residues observed in all scenarios after preliminary graphical analysis.

## Results and discussion

### Goodness-of-fit

Most of the best-ranked ANN presented a number of neurons on the hidden layer near the center of the tested amplitude (Table 1). This result indicates that the methodological approach used to define the number of the tested neurons was efficient.

**Table 1. Prediction and trend measures of the best ANN for the test dataset.**

| Variable | Scenario | ANN | Achitecture[a] | Hidden activation | Output activation | $r_{y\hat{y}}$ | RMSE (%) | MAE | bias | NRMSE (%) | WV |
|---|---|---|---|---|---|---|---|---|---|---|---|
| Cumulative cases | 1 | 3 | 159-22-2 | Exponential | Exponential | 0.9995 | 34.81 | 3.52 | 0.3960 | 4.3 | 9 |
| | 2 | 1 | 159-32-2 | Tanh | Logistic | 0.9997 | 17.08 | 4.67 | 0.9394 | 2.5 | 8 |
| | 3 | 3 | 159-27-2 | Logistic | Identity | 0.9995 | 20.26 | 6.70 | -0.0147 | 3.4 | 8 |
| | 4 | 4 | 159-32-2 | Tanh | Exponential | 0.9980 | 24.51 | 8.58 | -0.4841 | 6.4 | 9 |
| | 5 | 4[b] | 159-23-2 | Logistic | Tanh | 0.9994 | 19.77 | 11.45 | 1.3754 | 3.9 | 13 |
| | 6 | 5 | 159-29-2 | Logistic | Exponential | 0.9997 | 13.30 | 11.74 | -0.9107 | 2.6 | 5 |
| Cumulative deaths | 1 | 4 | 159-22-2 | Tanh | Exponential | 0.9992 | 110.33 | 0.51 | 0.1852 | 11.2 | 10 |
| | 2 | 1 | 159-32-2 | Tanh | Logistic | 0.9993 | 30.46 | 0.54 | 0.0291 | 3.7 | 11 |
| | 3 | 5 | 159-27-2 | Tanh | Identity | 0.9994 | 25.38 | 0.71 | -0.0267 | 3.4 | 8 |
| | 4 | 4 | 159-32-2 | Tanh | Exponential | 0.9984 | 39.55 | 0.86 | 0.1488 | 7.4 | 10 |
| | 5 | 1 | 159-27-2 | Logistic | Identity | 0.9993 | 26.38 | 1.17 | 0.0472 | 3.8 | 8 |
| | 6 | 2[b] | 159-25-2 | Tanh | Identity | 0.9994 | 24.24 | 1.24 | -0.1754 | 3.4 | 12.5 |
| Daily cases | 1 | 2 | 15-5-2 | Tanh | Exponential | 0.9614 | 51.97 | 19.60 | -0.2664 | 27.4 | 8 |
| | 2 | 4 | 15-5-2 | Tanh | Identity | 0.9582 | 50.56 | 17.30 | -0.6249 | 28.5 | 7 |
| | 3 | 2 | 15-8-2 | Exponential | Exponential | 0.9416 | 52.60 | 26.76 | 6.8013 | 34.0 | 10 |
| | 4 | 3 | 15-5-2 | Tanh | Logistic | 0.9363 | 55.07 | 31.13 | -14.0956 | 38.9 | 6 |
| | 5 | 1 | 15-10-2 | Tanh | Exponential | 0.9424 | 51.63 | 26.19 | -8.5063 | 34.3 | 9 |
| | 6 | 5 | 15-5-2 | Exponential | Exponential | 0.9285 | 51.71 | 27.85 | 0.6137 | 37.4 | 5 |
| Daily deaths | 1 | 5 | 15-11-2 | Logistic | Identity | 0.9834 | 57.19 | 1.15 | -0.1680 | 18.3 | 9.5 |
| | 2 | 3 | 15-10-2 | Logistic | Exponential | 0.9873 | 54.26 | 0.95 | -0.0654 | 15.8 | 7 |
| | 3 | 3 | 15-5-2 | Exponential | Exponential | 0.9817 | 52.34 | 1.03 | 0.0832 | 20.2 | 9 |
| | 4 | 4 | 15-15-2 | Exponential | Exponential | 0.9777 | 53.87 | 1.40 | 0.0097 | 22.5 | 5.5 |
| | 5 | 2 | 15-9-2 | Exponential | Exponential | 0.9806 | 49.60 | 1.04 | 0.1341 | 20.3 | 8 |
| | 6 | 4 | 15-15-2 | Exponential | Exponential | 0.9806 | 50.51 | 1.19 | 0.3022 | 20.4 | 6.5 |
| Standard hospital beds occupancy | 1 | 1 | 2-3-2 | Logistic | Tanh | 0.9978 | 24.05 | 15.63 | -7.1677 | 15.8 | 10 |
| | 2 | 4 | 2-8-2 | Logistic | Identity | 0.9985 | 7.61 | 7.41 | 2.4666 | 5.8 | 5 |
| | 3 | 2 | 2-7-2 | Logistic | Tanh | 0.9971 | 7.09 | 21.40 | 3.4817 | 8.1 | 8 |
| | 4 | 2 | 2-9-2 | Tanh | Identity | 0.9985 | 4.03 | 15.50 | 1.9142 | 5.3 | 6 |
| | 5 | 3 | 2-4-2 | Logistic | Identity | 0.9985 | 7.97 | 15.78 | 12.8058 | 6.8 | 8 |
| | 6 | 4 | 2-4-2 | Logistic | Identity | 0.9981 | 5.20 | 14.92 | 3.3560 | 5.9 | 6 |
| ICU beds occupancy | 1 | 4 | 2-6-2 | Tanh | Exponential | 0.9980 | 10.21 | 3.35 | -0.8465 | 6.4 | 7 |
| | 2 | 5 | 2-8-2 | Logistic | Exponential | 0.9985 | 11.04 | 4.52 | -1.5238 | 7.6 | 10 |
| | 3 | 1 | 2-9-2 | Tanh | Identity | 0.9982 | 5.97 | 8.26 | 3.8645 | 6.4 | 7 |
| | 4 | 1 | 2-3-2 | Logistic | Exponential | 0.9975 | 5.53 | 7.06 | 0.9004 | 6.9 | 5 |
| | 5 | 4 | 2-6-2 | Logistic | Logistic | 0.9983 | 7.78 | 7.94 | -0.1680 | 5.8 | 11 |
| | 6 | 1 | 2-8-2 | Logistic | Identity | 0.9991 | 4.05 | 4.81 | -0.7954 | 4.2 | 8 |

[a] represents the neuron number at input-hidden-output layers.

[b] the second-best ANN of the scenario.

For cumulative deaths and daily cases, the most frequent activation function in the hidden layer was the hyperbolic tangent, but for daily deaths, it was the exponential function. To cumulative cases, the function logistic was the most frequent, as well as to standard hospital beds and ICU beds. In the output layer, the predominant function was the exponential, followed by identity.

The correlation between the observed and forecasted values was higher than 0.99 to cumulative variables and hospital beds, and over 0.92 to daily cases and deaths. Only in scenario 5 of

daily death modeling, RMSE below 50% was observed when the daily variables were evaluated. For the daily cases and hospital beds, MAE over 11.74 and biases with greater amplitudes was observed. Lastly, only for the cumulative cases and ICU beds, NRMSE over 10% was not observed.

Several studies have used ANN and other techniques to model a dynamic temporal of cases and deaths caused by COVID-19 in the world [48–50]. Saba & Elsheikh reported $r_{y\hat{y}}$ like the cumulative variables of this study, after using nonlinear autoregressive ANN to model cumulative cases with data of 40 days [31]. Similarly, Torrealba-Rodriguez, Conde-Gutiérrez & Hernández-Javier, presented $r_{y\hat{y}}$ above 0.9 when modeling daily confirmed cases in Mexico by ANN [20].

Besides these, no other studies were found the presented measures of prediction accuracy like those used here for comparison, despite being an issue that can affect the forecasting accuracy. Furthermore, studies that modeled data on COVID-19 evaluating different scenarios in a seven-day variation were not found, which makes this study unique.

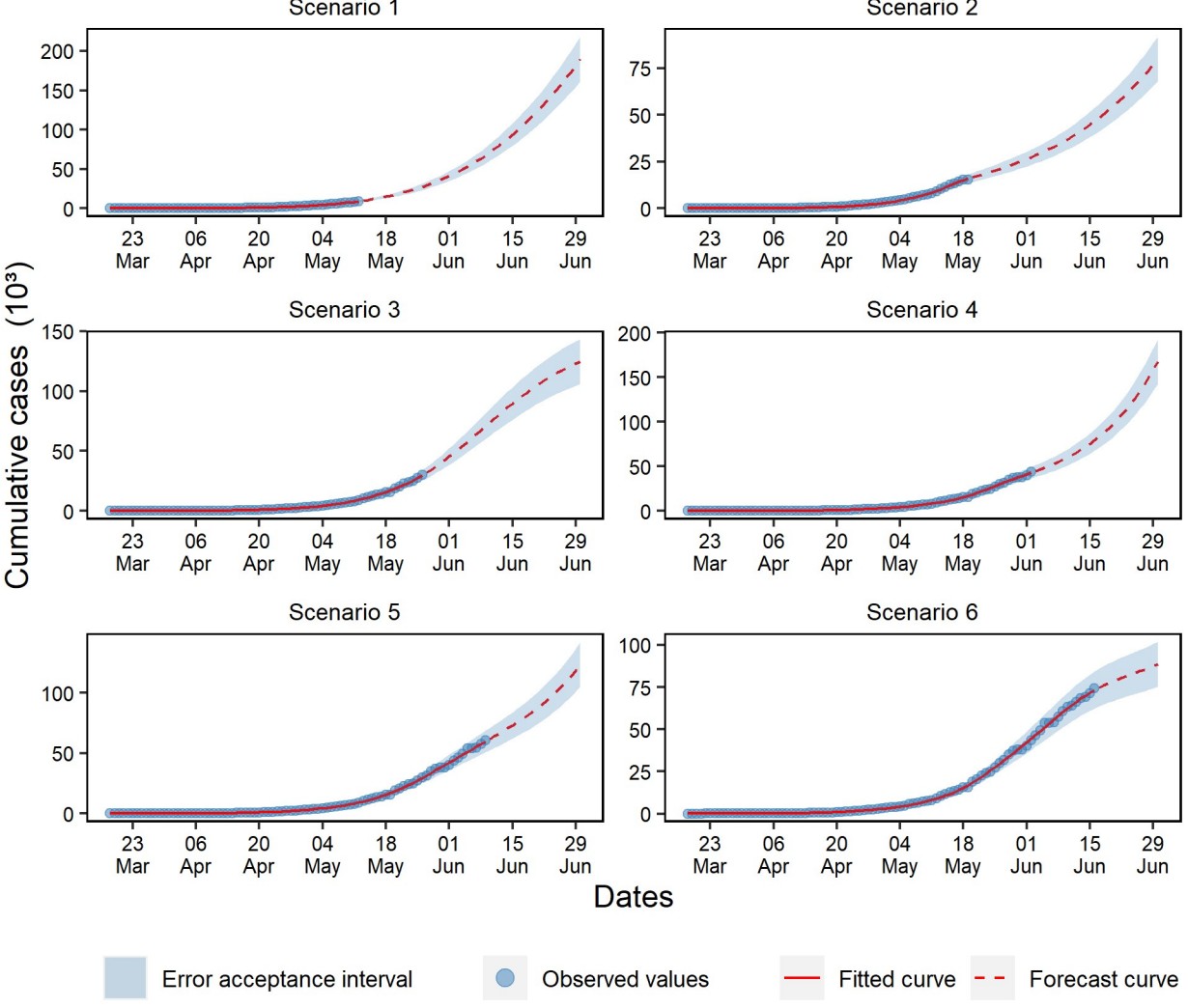

**Fig 4. Forecasting of cumulative cases in the six analyzed scenarios.**

### Forecasting analysis

The growth stage was identified from curve fitting. Then, to analyze the generated forecasts was presumed that the ANN captured the training data dynamics' signature and loaded the information to the cumulative (S4 Appendix), daily (S5 Appendix), and hospital beds (S6 Appendix) variables forecasts.

For cumulative cases, exponential growth was observed in scenarios 1, 2, and 4, mostly characterized by the exponential trend of the projected curve (Fig 4). In scenario 5, a growth slowdown was observed, which resulted in a linear growth phase. Even though scenarios 3 and 6 presented a plateau tendency, it is early to assume that the growth reached its peak. Therefore, it was necessary to evaluate the behavior of posterior data before making any assumptions.

An exponential growth profile of cumulative deaths was observed in the forecasts of scenarios 1 and 2 (Fig 5). In scenario 3, the forecast identified linear growth in the curve, and in

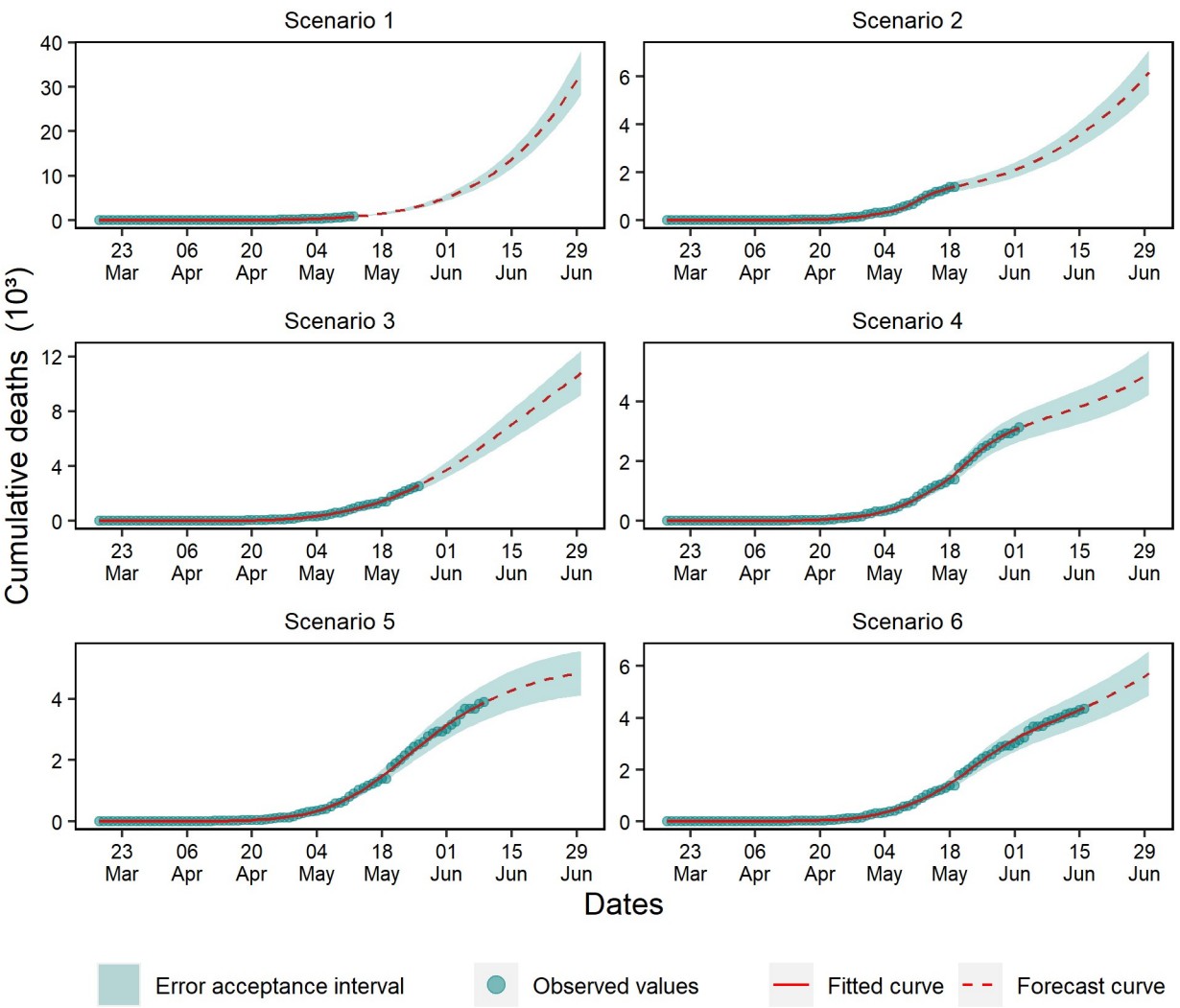

**Fig 5. Forecasting of cumulative deaths in the six analyzed scenarios.**

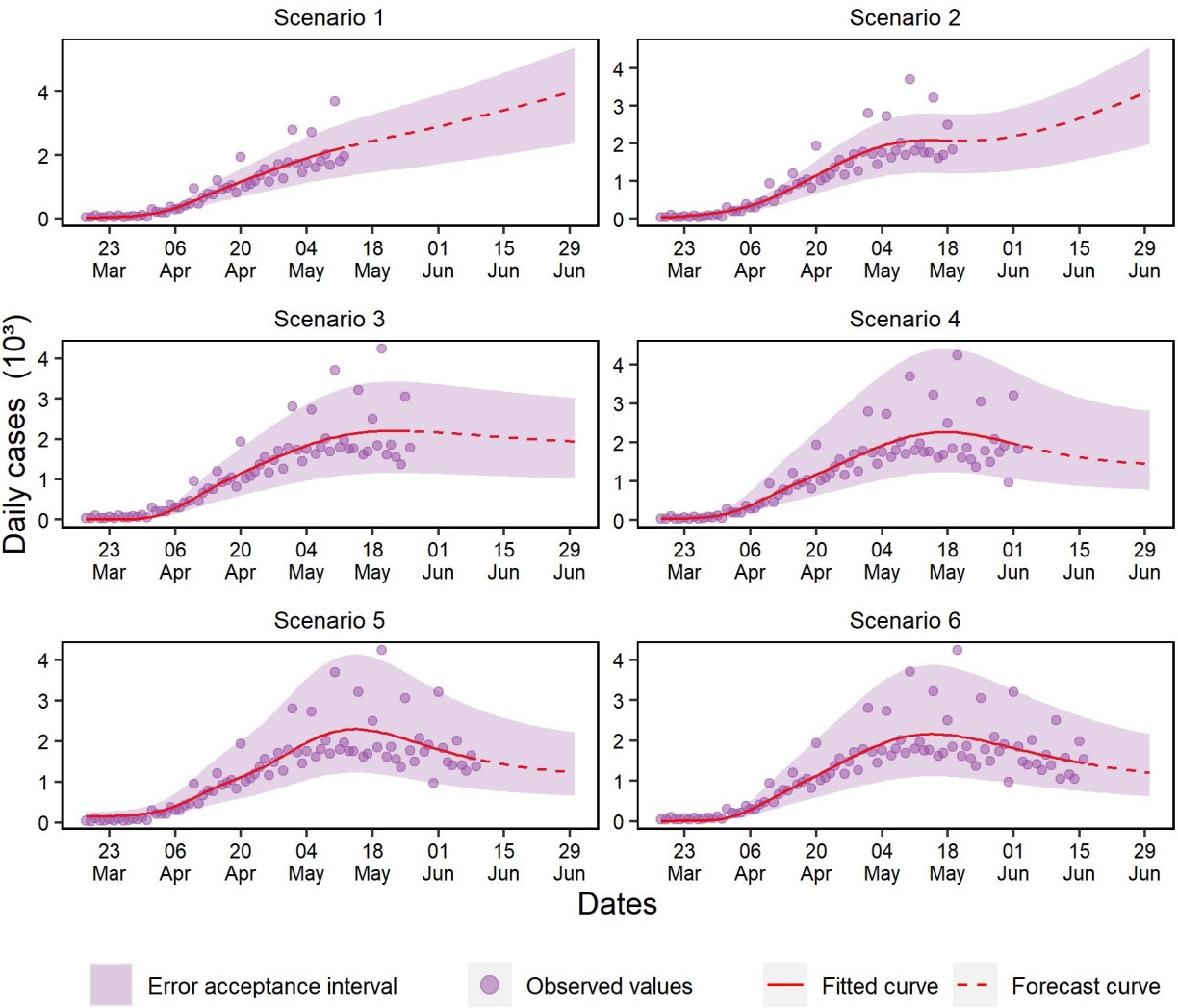

**Fig 6. Forecasting of daily cases for the six analyzed scenarios.**

scenario 5, there was a stabilization trend. Finally, scenarios 4 and 6 showed a wave effect due to the deceleration observed in the final data of the training set.

For daily cases, scenarios 1 and 2 forecasted data with exponential growth. However, scenario 2 presented a wave effect when capturing a possible increase of cases with the insertion of new data (Fig 6), while in scenario 3, a linear trend was observed—unlike the other scenarios. In other scenarios, a slight slowdown trend was observed, but it is premature to assume the epidemic is ending, mostly because the curve did not show an accentuated fall.

Only in daily deaths were observed data with slowdown trend (Fig 7). Scenarios 3 to 6 presented an accentuated fall and a decrease in the data variability. In scenario 1, the model forecasted growth continuity, while in scenario 2, the data oscillation captured a wave effect during the first peak that occurred between late April and mid-May.

To achieve stabilization, scenario 1 predicted that 700 standard hospital beds are necessary (Fig 8). This forecast increased linearly in scenario 2, while scenario 3 forecasted stabilization with approximately 800 standard hospital beds. In scenario 4, a wave effect was observed, and

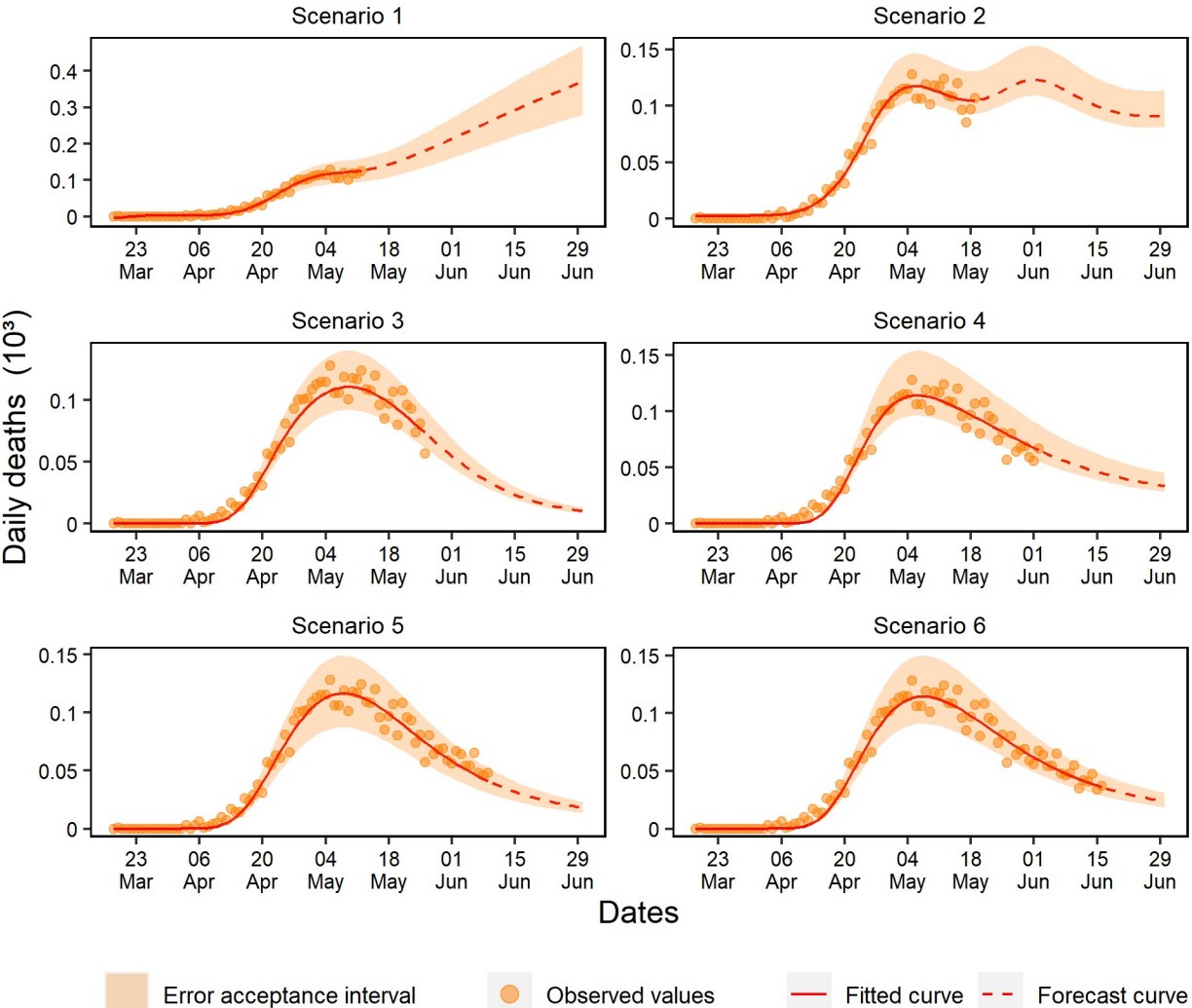

**Fig 7. Forecasting of daily deaths in the six analyzed scenarios.**

in scenarios 5 and 6 a decrease was observed followed by stabilization, probably because they captured a fall in the state's COVID-19 case growth.

On the one hand, scenario 1 (Fig 9) showed forecasts indicating overcrowding in ICU beds, probably due to skewed data. On the other hand, scenarios 2 and 3 present a growth behavior and the maximum capacity of ICU bed occupancy rates peaking on June 1st, which indicated the need for 500 beds. Scenario 4 predicted a linear growth, which stabilized in scenario 5 and then showed a declining trend in scenario 6.

Fundamentally, a forecast analysis can be described as the ability of a model to predict with accuracy. According to Hyndman & Athanasopoulos, predictability depends on how well the explanatory variables are understood, how much data is available, and if forecasting can affect what is trying to predict [51].

In the context of the COVID-19 pandemic, several studies are assuming that the data gathered about the disease is reliable and making predictions on how it is going to behave [35,52–55].

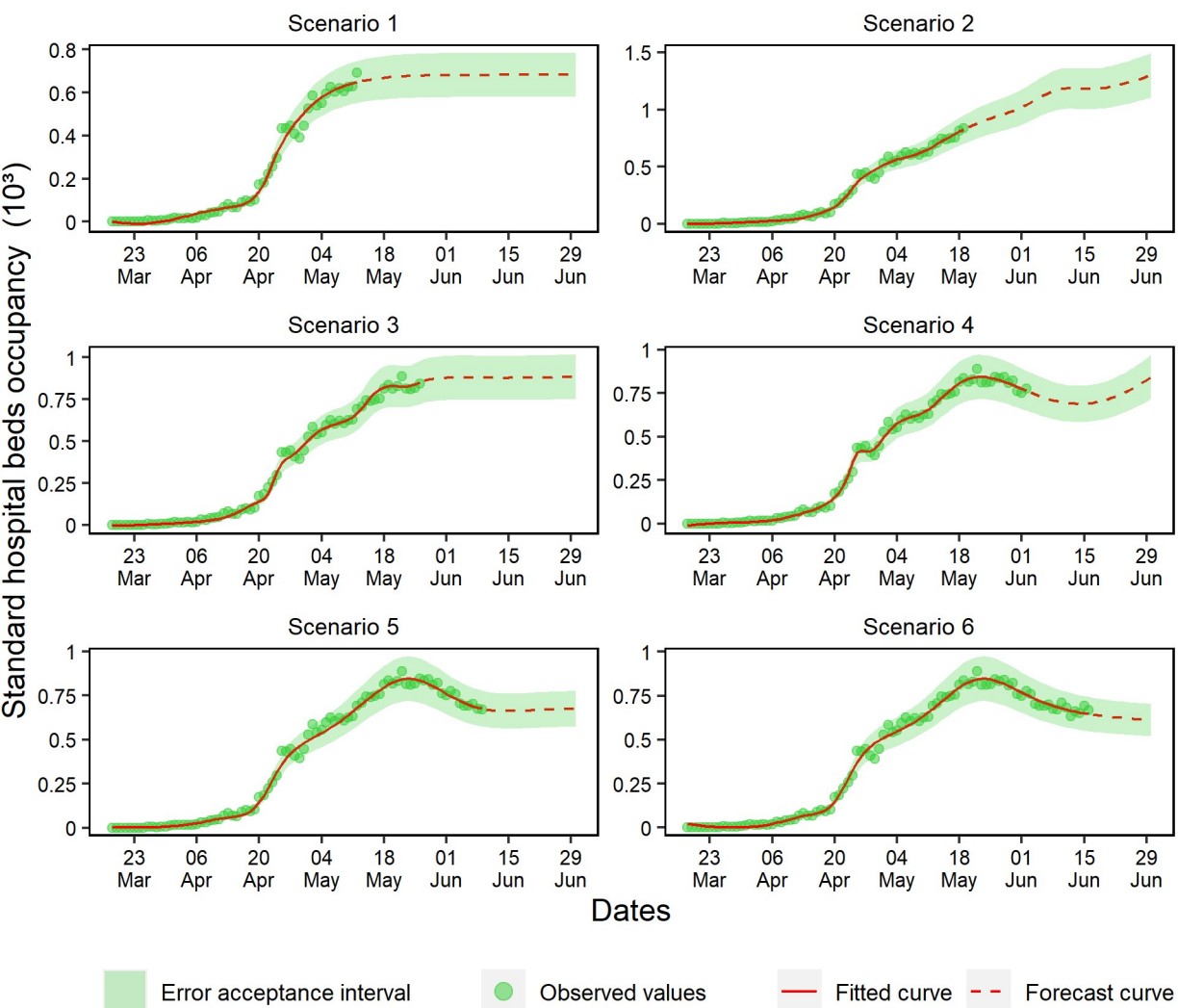

**Fig 8. Forecasting of standard hospital beds occupancy in the six analyzed scenarios.**

The application of forecasting techniques to prevent the short and long-term impacts of infectious diseases has been receiving more attention since the COVID-19 outbreak, especially regarding mortality and the health system's ability to care for all infected patients. Therefore, a better understanding of how the ANN captured the characteristics of historical data and used these pieces of information to create more reliable forecasts is necessary.

In their investigations, Fernandes and Petropoulos & Makridakis, analyzed COVID-19 forecasts in different moments since there is a high level of uncertainty regarding the published data in this pandemic [35,54]. Moreover, in this study, the forecasts were reassessed with new data added weekly.

Uncertainties are normally discussed in time series forecasting, and noisy data are one of the main sources. Chowell (2017) presented a noise quantification methodology to generate confidence intervals [15]. Fernandes (2020) discussed the insertion of new data and the progressive analysis of forecasting in several moments [35]. He assumed that the forecasting quality tends to improve and consequently decrease uncertainty associated with forecasts.

Therefore, the results presented in this research showed that the ANN generated forecasts with a trend more closely related to the observed data (Fig 2). This behavior became more

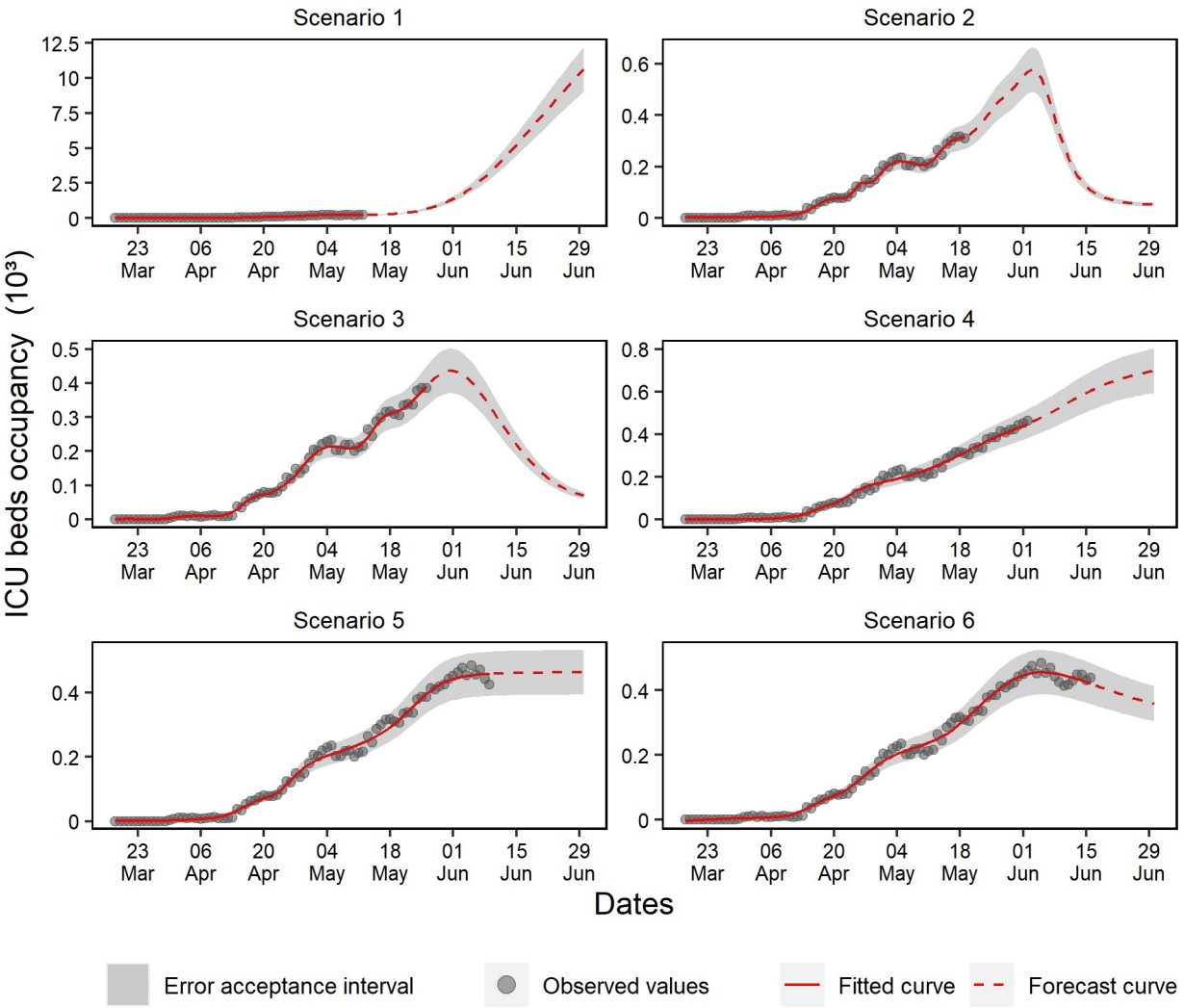

**Fig 9. Forecasting of ICU beds occupancy in the six analyzed scenarios.**

distinctive for the daily variables in scenarios 3, 4, 5, and 6, whose curves displayed a decreasing dispersion behavior as more data were inserted.

However, when exposed to the cumulative variables, the ANN proved to be sensitive, capturing the training data's oscillations and consequently loading these variations into the forecasts. Oscillations were observed on the weekends due to habitual underreported cases on these days.

**Evaluating forecast accuracy.** Due to the higher variability of data (Table 2), a loss of quality was observed in the prediction and trend measures compared to daily and cumulative variables. Regarding cumulative cases and deaths, scenario 5 stood out for presenting the best prediction and trend measures in the initial seven and fourteen days of forecasting.

Only one scenario for daily registered cases and deaths could not be highlight. The scenario 4 showed the lowest RMSE of daily cases forecasts for seven and fourteen days, but in this same variable, scenario 5 presented a pbias closer to zero.

**Table 2. Prediction and trend measures used to validate the forecasts.**

| Variable | Scenario | ANN | 7 days | | 14 days | |
|---|---|---|---|---|---|---|
| | | | RMSE (%) | pbias (%) | RMSE (%) | pbias (%) |
| Cumulative cases | 1 | 3 | 4.89 | -3.49 | 7.85 | -5.72 |
| | 2 | 1 | 23.44 | -21.07 | 32.98 | -27.89 |
| | 3 | 3 | 8.60 | 5.50 | 12.75 | 9.31 |
| | 4 | 4 | 5.63 | -5.19 | 5.01 | -1.91 |
| | 5 | 4 | 1.67 | 0.42 | 2.21 | 1.20 |
| | 6 | 5 | 7.73 | -7.03 | 12.33 | -10.78 |
| Cumulative deaths | 1 | 4 | 11.54 | 4.20 | 16.65 | 9.22 |
| | 2 | 1 | 27.51 | -25.89 | 30.63 | -28.81 |
| | 3 | 5 | 18.02 | 15.57 | 29.33 | 24.40 |
| | 4 | 4 | 8.43 | -7.78 | 9.81 | -9.24 |
| | 5 | 1 | 0.66 | -0.50 | 0.95 | -0.76 |
| | 6 | 2 | 4.02 | 3.68 | 8.98 | 7.50 |
| Daily cases | 1 | 2 | 31.32 | 22.89 | 41.73 | 30.00 |
| | 2 | 4 | 44.64 | 8.69 | 41.28 | 18.14 |
| | 3 | 2 | 36.74 | 28.31 | 39.45 | 35.58 |
| | 4 | 3 | 25.77 | 23.32 | 29.46 | 20.63 |
| | 5 | 1 | 32.13 | 4.33 | 30.15 | 10.70 |
| | 6 | 5 | 26.33 | 18.00 | 34.51 | 32.09 |
| Daily deaths | 1 | 5 | 35.64 | 34.55 | 69.66 | 70.21 |
| | 2 | 3 | 37.46 | 36.27 | 60.29 | 60.95 |
| | 3 | 3 | 11.64 | -7.32 | 18.56 | -15.26 |
| | 4 | 4 | 13.17 | 9.86 | 17.34 | 14.83 |
| | 5 | 2 | 21.57 | -14.75 | 25.80 | -16.49 |
| | 6 | 4 | 22.71 | -0.56 | 23.21 | 7.88 |
| Standard hospital beds occupancy | 1 | 1 | 13.84 | -12.78 | 16.75 | -15.64 |
| | 2 | 4 | 9.06 | 7.71 | 18.80 | 16.14 |
| | 3 | 2 | 10.51 | 9.52 | 18.97 | 17.77 |
| | 4 | 2 | 4.59 | 4.12 | 4.62 | 3.68 |
| | 5 | 3 | 3.62 | -0.79 | 6.44 | 3.34 |
| | 6 | 4 | 4.03 | 2.21 | 8.82 | 7.21 |
| ICU beds occupancy | 1 | 4 | 15.81 | -14.59 | 33.48 | 8.99 |
| | 2 | 5 | 9.18 | 7.70 | 16.02 | 13.12 |
| | 3 | 1 | 3.26 | -1.06 | 12.34 | -8.58 |
| | 4 | 1 | 11.07 | 7.68 | 23.60 | 19.87 |
| | 5 | 4 | 6.80 | 6.30 | 4.86 | 3.95 |
| | 6 | 1 | 11.74 | -11.51 | 11.42 | -11.14 |

Scenario 4 also presented the lowest RMSE in the fourteen-day forecast of daily deaths, while scenario 3 achieved the same results in the seven-day forecast. Apart from that, the lowest pbias were observed in the scenario 6 to for seven- and fourteen-day forecasts. Similarly, none of the scenarios were completely accurate in the standard hospital beds and ICU beds forecasts.

However, just like the cumulative cases and deaths forecast, scenario 5 generated the most accurate standard hospital beds and ICU beds forecasts for seven and fourteen days. In the fourteen-day forecast for standard hospital beds, this scenario showed the lowest pbias, while

scenario 4 obtained the highest RMSE. Finally, scenario 3 was the most accurate when forecasting ICU beds for seven days.

From these prediction and trend measures, it was possible to infer that there is no direct relationship between the quantity of data used for training of the ANNs and the forecast accuracy for seven and fourteen days. Excluding scenario 2, the ANN technique has shown itself capable of forecasting cumulative cases and deaths for seven days with an RMSE below 18.02%.

With ANN assistance, Moftakhar, Seif & Safe used 35-day old data to model the number of COVID-19 new daily cases in Iran [56]. They performed forecasts for six days with an underestimation bias and pbias equal to -53.5%, much higher than the pbias of scenario 3—the highest scenario obtained in this study. This may have been motivated by fewer days used for training or by the used ANN hyperparameters.

Eshragh et al. evaluated the quality of the forecasts by identifying two rupture points [57]. Therefore, they were able to determine the initial, intermediate, and final phases, and then evaluated the accuracy of forecasts using mean absolute percentage error.

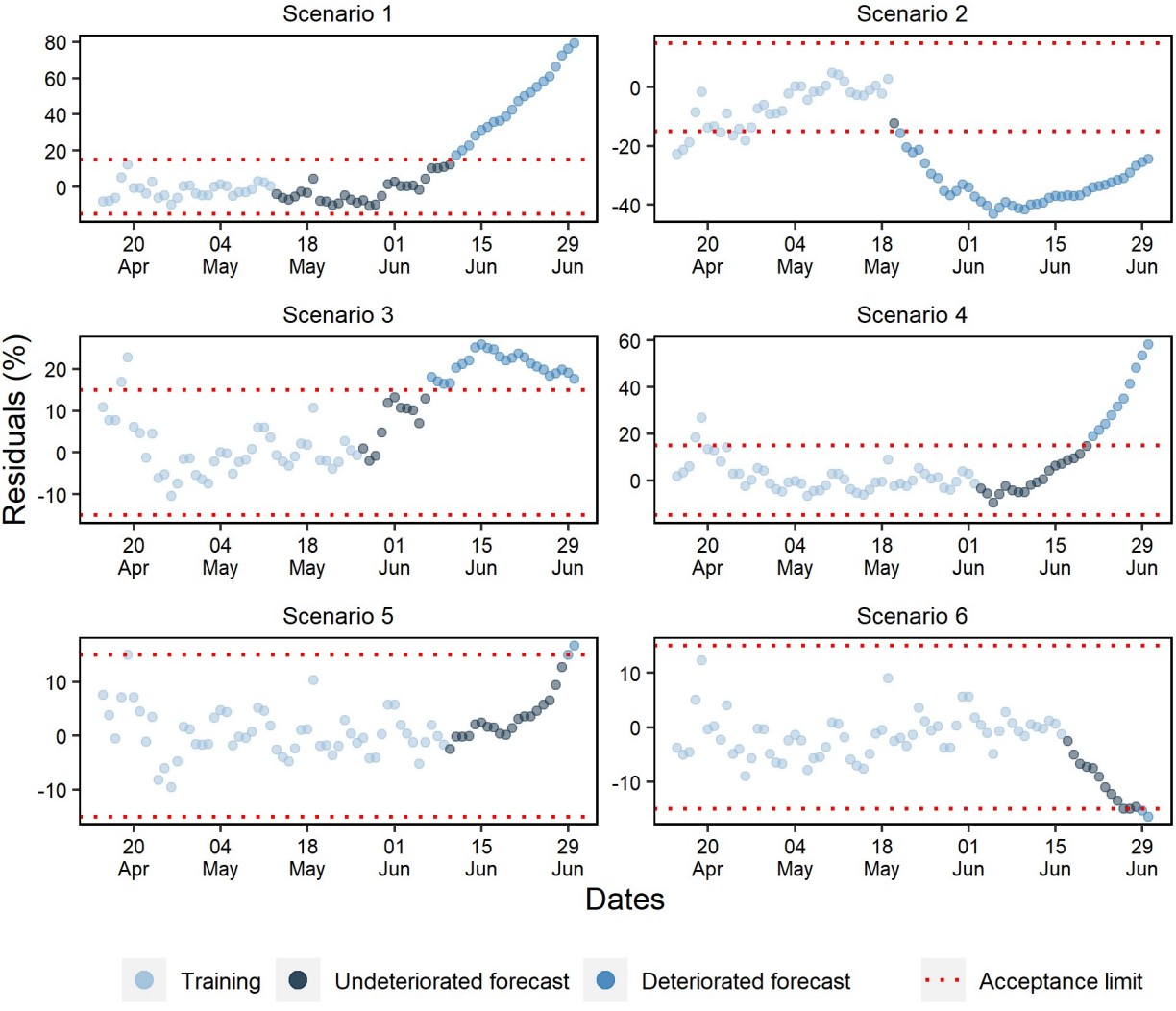

**Fig 10. Percentual residuals for cumulative cases of the 6 analyzed scenarios.**

There is a large discussion on assessing forecast quality through prediction and trend measures in the literature. For instance, Hyndman & Koehler compared prediction measures for univariate time series forecasts and found problems related to division by zero and prediction trends with high values [58].

However, for epidemic models, division by zero does not occur due to the curve growth feature and the lack of zero notifications on daily variables data. This study focused on short-term time horizons and generated low-value results for the measure values. For this reason, it is reasonable to say that the higher values indicated early deterioration when compared to other values in the same variable.

### Forecast deterioration analysis

While evaluating the ANN training periods of the cumulative cases, a random pattern was identified in the percentage residuals (Fig 10). Except for scenario 2, where a tendency to underestimate percentage residuals and a rapid deterioration of forecasts was observed, the

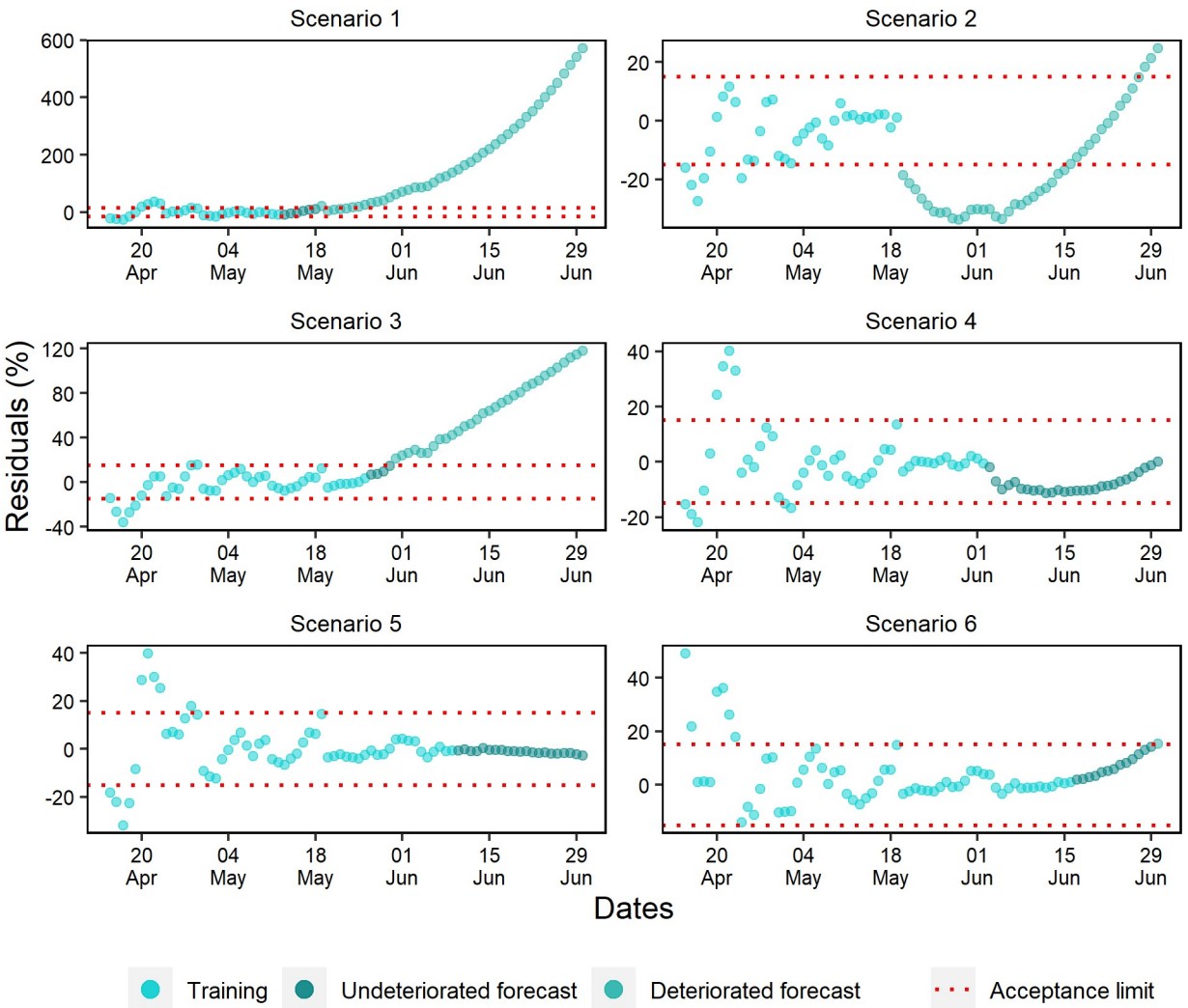

**Fig 11. Percentual residuals for cumulative deaths of the 6 analyzed scenarios.**

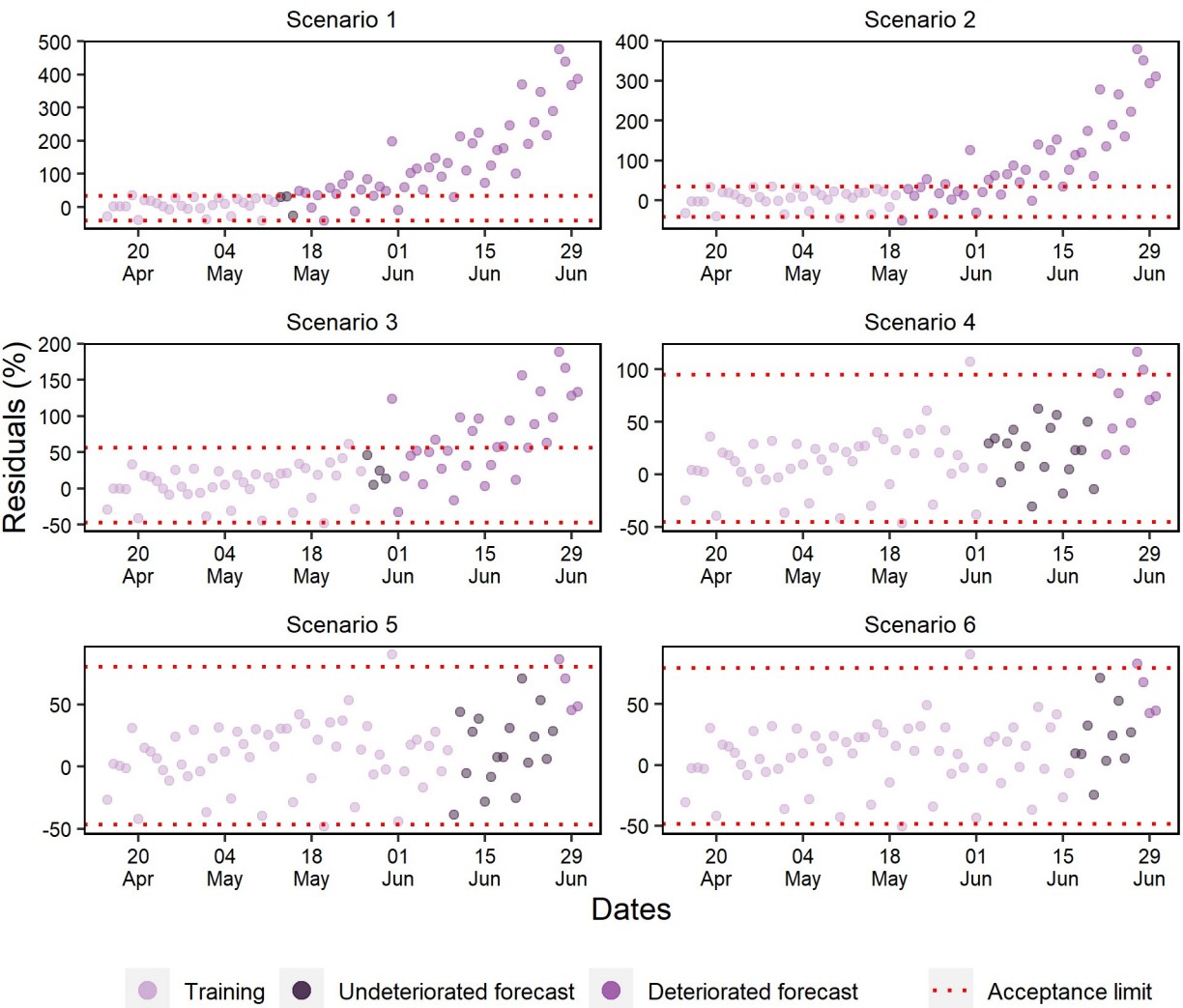

**Fig 12. Percentual residuals for daily cases of the 6 analyzed scenarios.**

other scenarios projected without deteriorating beyond 11 days. Furthermore, as more data was added, it was noted that the forecast residuals were more stable within the 15% range.

Consistent and stable predictions were observed when entering new ANN training data for cumulative deaths (Fig 11). On one hand, scenarios 1, 2, and 3 presented greater variability of residuals in the fit and therefore showed biased forecasts. On the other hand, scenarios 4, 5, and 6 confirmed the ANNs ability to perform forecasting within the range ±15% with accuracy for at least 13 days.

Well-fitted trainings were observed in all daily case scenarios corroborated by the occurrence of a random pattern for the last 28 days of ANN training (Fig 12). However, only in scenarios 4, 5, and 6 were forecasts obtained with more than 10 days without deterioration.

The last 28 days of the training set for daily deaths showed percentual residuals with a random pattern and approximately constant variation (Fig 13). However, scenarios 1 and 2 presented fast deterioration of forecasts due to the early exponential growth.

Scenarios 3, 4, and 6 produced better forecasts, but unlike the others, scenario 3 was the only one that presented a strong underestimation tendency. At last, early deterioration was

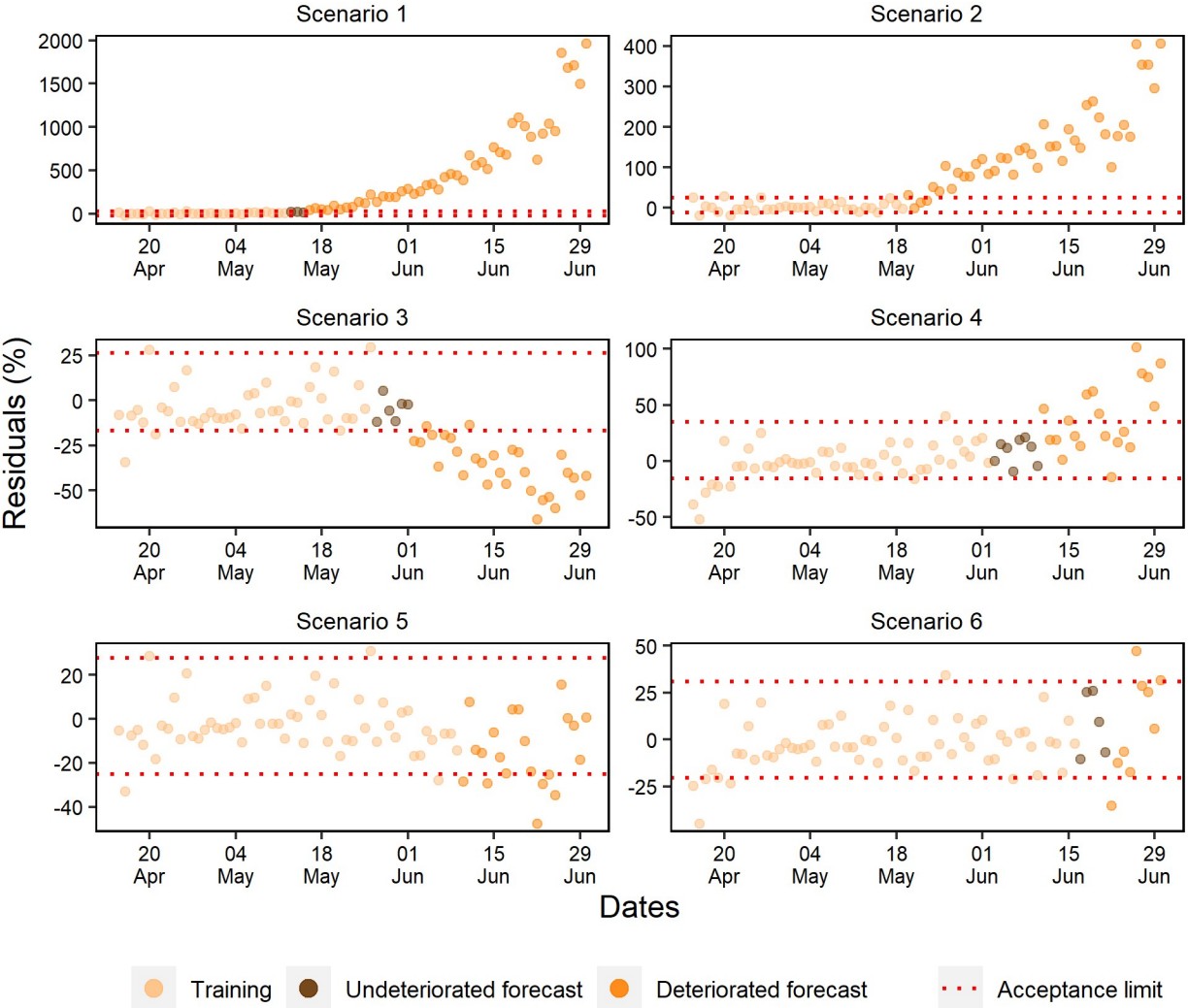

**Fig 13. Percentual residuals for daily deaths of the 6 analyzed scenarios.**

observed in scenario 5 due to the acceptance criteria. In general, a significant part of the subsequent data of this scenario remained within the acceptance range, with a slight trend of underestimation.

Scenarios 1, 2, and 3 forecasted standard hospital beds occupancy without deterioration for a maximum of respectively 5, 9, and 4 days (Fig 14). Scenario 4 consistently forecasted for 18 days, and scenarios 6 and 5 validated the ANN capacity of forecasting within the range of ±15% for respectively 10 and 13 days.

The forecasting of scenarios 1, 2, and 4 deteriorated up to 6 days for the ICU beds occupancy (Fig 15). Scenario 1 deteriorated from the first-day forecast; however, scenarios 3, 5, and 6 forecasted respectively for 9, 20, and 13 days.

The percentual residuals of the daily cases forecasts and deaths corroborated with the accuracy measures of the validation set because they were higher regarding the measures of cumulative deaths and cases. This situation was expected due to these daily variables presenting a higher variability.

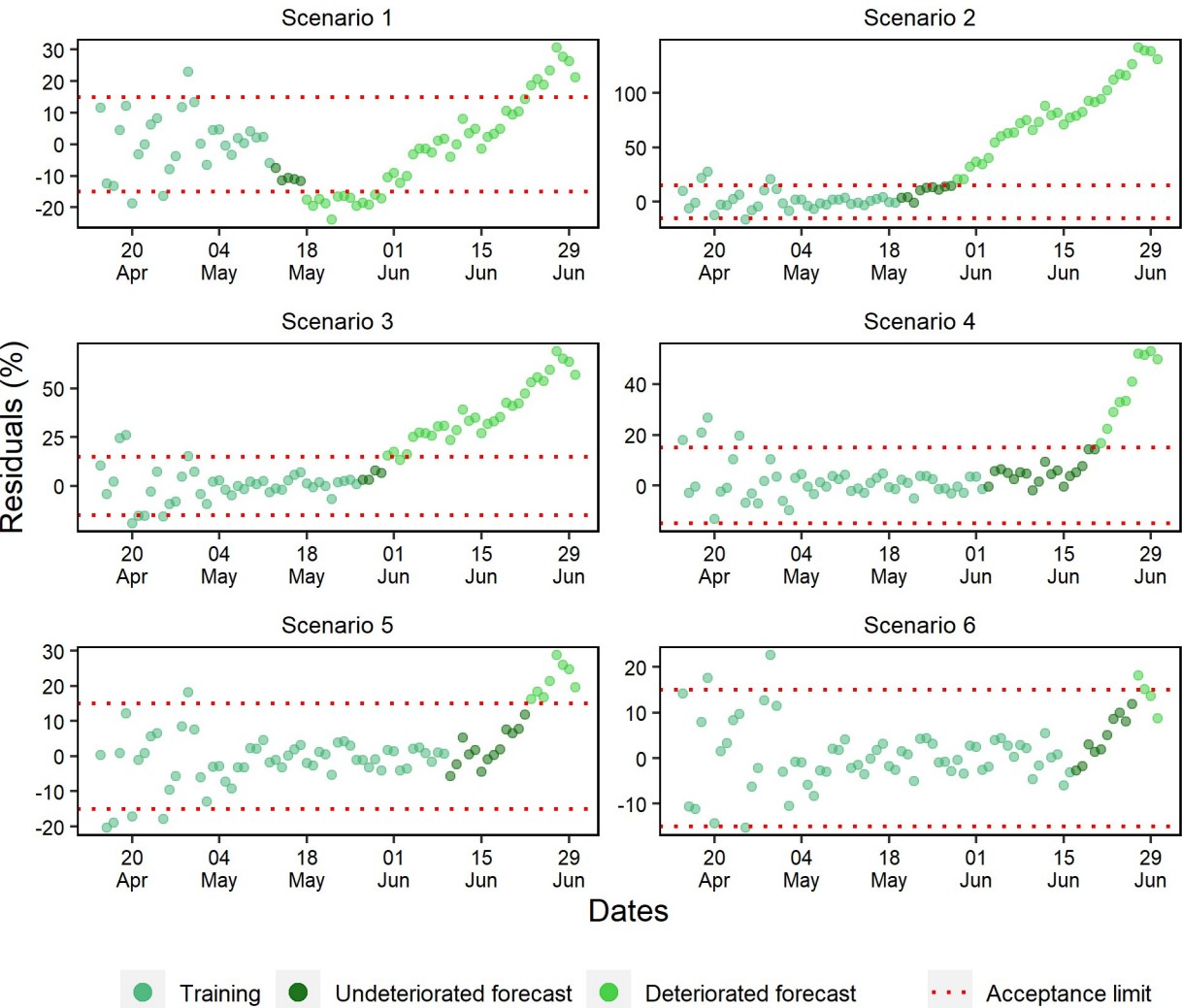

**Fig 14. Percentual residuals for standard hospital beds occupancy of the 6 analyzed scenarios.**

The diagnosis of residuals is constantly described in the literature, especially when it comes to time series. According to Athanasopoulos and Hyndman, a good forecasting method generates residuals with a mean equal to zero and uncorrelated. Therefore, these residuals are useful to verify if the technique has properly collected the information from the data [51].

The epidemic data are characterized for initial exponential growth, and after modeling, they tend to present residuals that do not follow a random pattern, suggesting a bad fit to the data. However, random pattern was collected in the last 28 days of the ANN training period in most scenarios.

The forecast deterioration analysis is usually conducted in residual graphs or by accuracy and trend measurements [59,60]. However, the acceptance criteria are often based on statistical significance.

Mestre suggests that it is possible to consider statistically acceptable forecasts even if they do not present errors with normal distribution, but they need to be random and present constant variance [61]. In this sense, Burnham, Anderson & Huyvaert stated that there are no better criteria to be applied due to predictive analytics' underlying uncertainties [62].

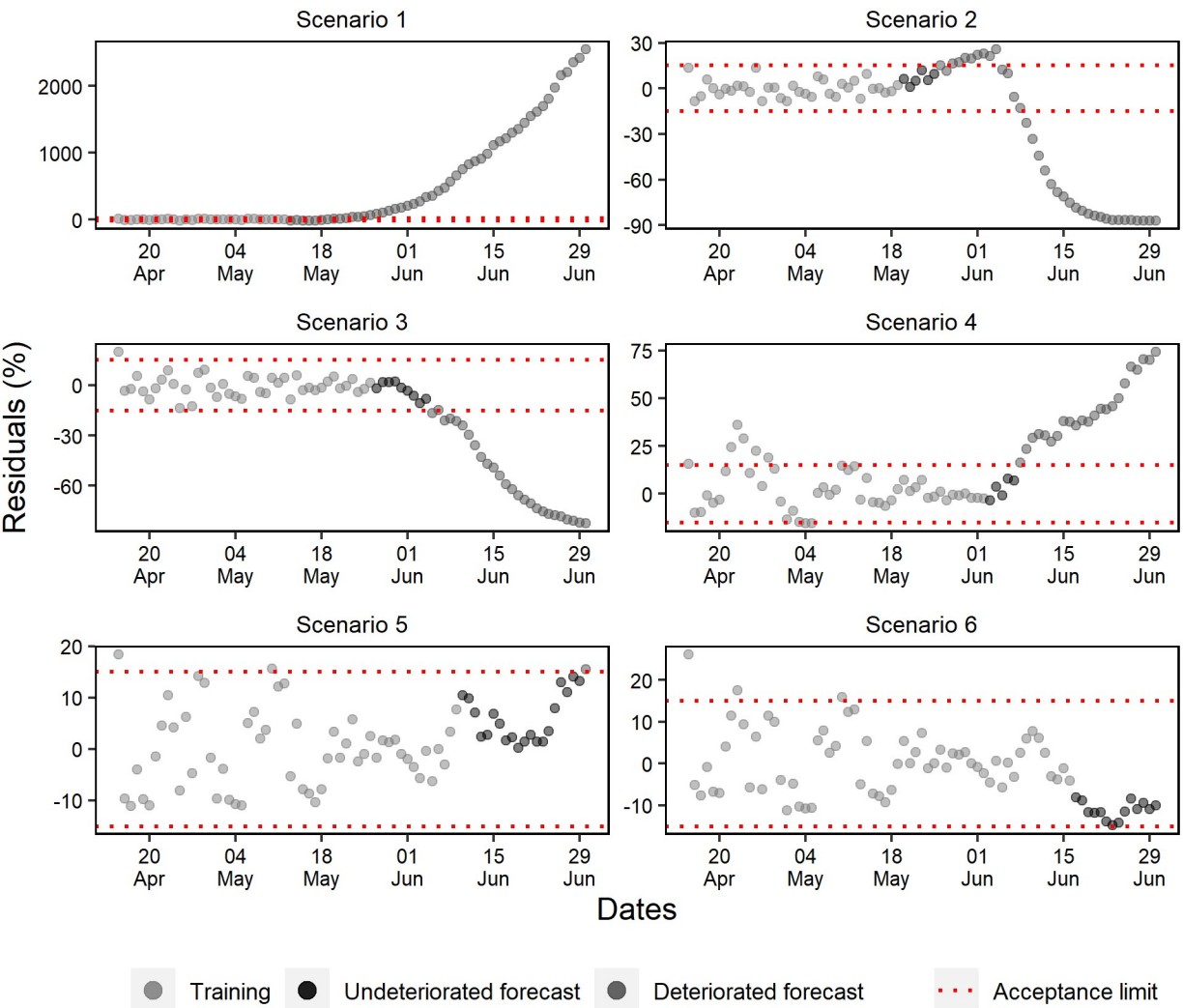

**Fig 15. Percentual residuals for ICU bed occupancy in the 6 analyzed scenarios.**

Few variability in the residual graphic of the cumulative variables was observed. For this reason, the establishment of more restricted forecast acceptance criteria was allowed. For daily variables, it was necessary to use a broader acceptance strategy of the residuals, due to the high variability observed in these variables. Therefore, accepting forecasts that present residuals like those observed in the prediction of the training set was an acceptable criterion.

## Conclusions

This study presented forecasts of the number of COVID-19 confirmed cases, deaths, and demands for hospital beds in the state of Pará, in the Brazilian Amazon, from ANN. As a result, it was attempted to identify whether the quality of forecasts increases as new data is inserted, and on how many days the forecasting starts to lose its validity and deteriorate. The results show that the ANNs generate forecasts that tended to be closer to the observed data in the daily variables and hospital beds as new data are inserted in the ANN training dataset.

Nevertheless, there is no direct relationship between the amount of data used for ANN training and the accuracy and trend measures of the 7 and 14-day forecasts. For, when forecasting for 7 and 14 days, except for daily cases, scenario 6 does not present better accuracy measures than scenario 5. However, in this variable, scenario 5 does not show a more precise forecast than scenario 4.

Unlike scenario 2, the other cumulative case scenarios forecast at least 11 days without deteriorating. For cumulative deaths, scenario 2 deteriorates on the first day; however, the forecasts of scenarios 4, 5, and 6 do not deteriorate within less than 13 days.

Daily cases of scenarios 4, 5, and 6 deteriorate within more than 10 days. As for daily deaths, only scenario 4 forecasts beyond 7 days without deterioration, and unlike the other variables, scenario 5 does not forecast any day without deteriorating.

Only scenarios 1 and 3 forecasted standard hospital beds occupancy without deteriorating under 9 days. Similarly, scenarios 3, 4, and 6 also forecasted ICU beds occupancy with at least 9 days without deterioration.

In summary, the artificial intelligence technique used in this study can assist government authorities to reduce the impacts of the pandemic. This technique can forecast in advance the number of potential cases, deaths, and hospital beds occupancy with a low error percentual.

## Supporting information

**S1 Appendix. Data set with input and output variables used for training and predictions of cumulative variables made by the five best ANNs.**
(XLSX)

**S2 Appendix. Data set with input and output variables used for training and predictions of daily variables made by the five best ANNs.**
(XLSX)

**S3 Appendix. Data set with input and output variables used for training and predictions of the hospital beds occupancy made by the five best ANNs.**
(XLSX)

**S4 Appendix. Data set with forecasts of the cumulative variables made by the five best ANNs.**
(XLSX)

**S5 Appendix. Data set with forecasts of the daily variables made by the five best ANNs.**
(XLSX)

**S6 Appendix. Data set with forecasts of the hospital beds occupancy made by the five best ANNs.**
(XLSX)

**S1 File.**
(PDF)

## Acknowledgments

To the members of the Voluntary translation of informative materials related to the COVID-19 project, offered by NUPEL / UFBA and supervised by professors M. Daniel Vasconcelos B. Oliveira, Dr. Feibriss Henrique Meneghelli Cassilhas, Dr. Lucielen Porfirio, and Dr. Monique Pfau. Translators: Dara Lanzotti, Luciana Cupertino, Maria Boaventura, and Nathália Borges.

To State of Pará Government for making the data available. To Universidade Federal Rural da Amazônia for making computers available for processing the developed algorithms.

## Author Contributions

**Conceptualization:** Marcus de Barros Braga, Rafael da Silva Fernandes, Gilberto Nerino de Souza, Jr, Jonas Elias Castro da Rocha, Antonio Carlos Rosário Vallinoto.

**Data curation:** Marcus de Barros Braga, Rafael da Silva Fernandes, Jonas Elias Castro da Rocha, Cícero Jorge Fonseca Dolácio, Ivaldo da Silva Tavares, Jr, Rommel Thiago Jucá Ramos, Marcel do Nascimento Botelho, Antonio Carlos Rosário Vallinoto.

**Formal analysis:** Marcus de Barros Braga, Rafael da Silva Fernandes, Gilberto Nerino de Souza, Jr, Cícero Jorge Fonseca Dolácio, Luana Lorena Silva Rodrigues, Adriana Ribeiro Carneiro, Silvana Rossy de Brito, Hugo Alex Carneiro Diniz, Antonio Carlos Rosário Vallinoto.

**Investigation:** Rafael da Silva Fernandes, Cícero Jorge Fonseca Dolácio, Luana Lorena Silva Rodrigues, Adriana Ribeiro Carneiro, Marcel do Nascimento Botelho, Antonio Carlos Rosário Vallinoto.

**Methodology:** Marcus de Barros Braga, Rafael da Silva Fernandes, Gilberto Nerino de Souza, Jr, Jonas Elias Castro da Rocha, Cícero Jorge Fonseca Dolácio, Ivaldo da Silva Tavares, Jr.

**Project administration:** Marcus de Barros Braga, Jonas Elias Castro da Rocha.

**Resources:** Luana Lorena Silva Rodrigues.

**Software:** Cícero Jorge Fonseca Dolácio, Ivaldo da Silva Tavares, Jr, Fernando Napoleão Noronha, Rommel Thiago Jucá Ramos.

**Supervision:** Marcus de Barros Braga, Gilberto Nerino de Souza, Jr, Jonas Elias Castro da Rocha, Rommel Thiago Jucá Ramos, Silvana Rossy de Brito, Hugo Alex Carneiro Diniz, Marcel do Nascimento Botelho, Antonio Carlos Rosário Vallinoto.

**Validation:** Marcus de Barros Braga, Rafael da Silva Fernandes, Gilberto Nerino de Souza, Jr, Jonas Elias Castro da Rocha, Cícero Jorge Fonseca Dolácio, Ivaldo da Silva Tavares, Jr, Raphael Rodrigues Pinheiro, Luana Lorena Silva Rodrigues, Rommel Thiago Jucá Ramos, Adriana Ribeiro Carneiro, Silvana Rossy de Brito, Hugo Alex Carneiro Diniz, Marcel do Nascimento Botelho, Antonio Carlos Rosário Vallinoto.

**Visualization:** Rafael da Silva Fernandes, Gilberto Nerino de Souza, Jr, Jonas Elias Castro da Rocha, Cícero Jorge Fonseca Dolácio, Ivaldo da Silva Tavares, Jr, Raphael Rodrigues Pinheiro, Fernando Napoleão Noronha, Rommel Thiago Jucá Ramos, Hugo Alex Carneiro Diniz.

**Writing – original draft:** Marcus de Barros Braga, Rafael da Silva Fernandes, Gilberto Nerino de Souza, Jr, Cícero Jorge Fonseca Dolácio, Ivaldo da Silva Tavares, Jr, Adriana Ribeiro Carneiro, Antonio Carlos Rosário Vallinoto.

**Writing – review & editing:** Marcus de Barros Braga, Rafael da Silva Fernandes, Cícero Jorge Fonseca Dolácio, Luana Lorena Silva Rodrigues, Antonio Carlos Rosário Vallinoto.

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
