## [Decision Letter · Decision Letter 0]

12 Jan 2021

PONE-D-20-38808

Artificial neural networks for short-term forecasting of cases, deaths, and hospital beds occupancy in the COVID-19 pandemic at the Brazilian Amazon

PLOS ONE

Dear Dr. Braga,

Thank you for submitting your manuscript to PLOS ONE. After careful consideration, we feel that it has merit but does not fully meet PLOS ONE’s publication criteria as it currently stands. Therefore, we invite you to submit a revised version of the manuscript that addresses the points raised during the review process.

We look forward to receiving your revised manuscript.

Kind regards,

Qiang Zeng, Ph.D.

Academic Editor

PLOS ONE

Journal Requirements:

3. We note that Figure 1 in your submission contain map images which may be copyrighted. All PLOS content is published under the Creative Commons Attribution License (CC BY 4.0), which means that the manuscript, images, and Supporting Information files will be freely available online, and any third party is permitted to access, download, copy, distribute, and use these materials in any way, even commercially, with proper attribution. For these reasons, we cannot publish previously copyrighted maps or satellite images created using proprietary data, such as Google software (Google Maps, Street View, and Earth). For more information, see our copyright guidelines: http://journals.plos.org/plosone/s/licenses-and-copyright.

3.1.    You may seek permission from the original copyright holder of Figure 1 to publish the content specifically under the CC BY 4.0 license. 

3.2.    If you are unable to obtain permission from the original copyright holder to publish these figures under the CC BY 4.0 license or if the copyright holder’s requirements are incompatible with the CC BY 4.0 license, please either i) remove the figure or ii) supply a replacement figure that complies with the CC BY 4.0 license. Please check copyright information on all replacement figures and update the figure caption with source information. If applicable, please specify in the figure caption text when a figure is similar but not identical to the original image and is therefore for illustrative purposes only.

Reviewers' comments:

Reviewer's Responses to Questions

**Comments to the Author**

1. Is the manuscript technically sound, and do the data support the conclusions?

Reviewer #1: Yes

Reviewer #2: Yes

2. Has the statistical analysis been performed appropriately and rigorously? 

Reviewer #1: N/A

Reviewer #2: Yes

3. Have the authors made all data underlying the findings in their manuscript fully available?

Reviewer #1: Yes

Reviewer #2: Yes

4. Is the manuscript presented in an intelligible fashion and written in standard English?

Reviewer #1: Yes

Reviewer #2: Yes

5. Review Comments to the Author

Reviewer #1: This study applies ANNs for short-term forecasting of cases, deaths, and hospital beds occupancy in the COVID-19 pandemic at the Brazilian Amazon. The research topic is worth of investigation. The methods sound reasonable. Some minor revisions are needed before its publication in PLOS ONE. Specifically,

More illustration on the strength of ANNs are expected in the Introduction section.

Additionally, more references on the application of ANNs and the used criteria for measuring goodness-of-fit should be acknowledged, such as:

Modeling nonlinear relationship between crash frequency by severity and contributing factors by neural networks. Analytic Methods in Accident Research, 2016, 10: 12-25.

Macro and micro models for zonal crash prediction with application in hot zones identification. Journal of Transport Geography, 2016, 54: 248-256.

Bayesian spatial-temporal model for the main and interaction effects of roadway and weather characteristics on freeway crash incidence. Accident Analysis and Prevention, 2019, 132, 105249.

In the ANN modeling section, a figure showing how the output and inputs connected would present the model more explicitly.

The Broyden-Fletcher-Goldfarb-Shanno iterative algorithm used for ANN training is rare in the previous studies. A detailed description of this algorithm is anticipated.

It would be interesting and further demonstrate the advantages of ANNs to compare their performance with that of time series models that are used in the previous studies on forecasting COVID-19 cases.

Reviewer #2: This study presents an approach based on artificial neural networks for the daily and cumulative forecasts of cases and deaths caused by COVID-19, and the forecast of demand for hospital beds. The topic is important. The methods sound. The results are meaningful. It is suggested to be published.

6. PLOS authors have the option to publish the peer review history of their article (what does this mean?). If published, this will include your full peer review and any attached files.

Reviewer #1: No

Reviewer #2: No

---

## [Author Response · Author response to Decision Letter 0]

8 Feb 2021

Response to Reviewers

Journal Requirements: 

and 

RESPONSE: We have reviewed and edited the manuscript according to the journal's recommendations.

RESPONSE: According to reviewers the manuscript was presented in an intelligible fashion and written in standard English. We have submitted the manuscript to an official translation service at the Federal University of Bahia, Brazil. We will send the translation declaration as an attached file. 

3. We note that Figure 1 in your submission contain map images which may be copyrighted. All PLOS content is published under the Creative Commons Attribution License (CC BY 4.0), which means that the manuscript, images, and Supporting Information files will be freely available online, and any third party is permitted to access, download, copy, distribute, and use these materials in any way, even commercially, with proper attribution. For these reasons, we cannot publish previously copyrighted maps or satellite images created using proprietary data, such as Google software (Google Maps, Street View, and Earth). For more information, see our copyright guidelines: http://journals.plos.org/plosone/s/licenses-and-copyright. We require you to either (1) present written permission from the copyright holder to publish these figures specifically under the CC BY 4.0 license, or (2) remove the figures from your submission:

RESPONSE: Thank you for the careful revision, but the Figure 1 was created by the manuscript's authors and they own its use rights. 

Review Comments to the Author - Reviewer #1:

1 - More illustration on the strength of ANNs are expected in the Introduction section 

RESPONSE: Thank you for the suggestion. We have added in the introduction section the following text (line 70), reinforcing the strength of the ANNs:

“The main characteristic of ANN is self-learning without prior knowledge of the complex non-linear relationships that exist between the input and output variables [22]. This is due to the massive and parallel processing of neurons and the tolerance to noise [23]. In addition, this technique captures small distortions in the observed data and transfers them for projections differently from mechanistic models [24]. Another advantage is that this type of approach also makes it possible to use several predictor variables simultaneously, such as demographic data and incidence curves, which helps in capturing the dynamics of virus transmission in the cities over time [25,26].”

2 - Additionally, more references on the application of ANNs and the used criteria for measuring goodness-of-fit should be acknowledged, such as: 

 Modeling nonlinear relationship between crash frequency by severity and contributing factors by neural networks. Analytic Methods in Accident Research, 2016, 10: 12-25.

 Macro and micro models for zonal crash prediction with application in hot zones identification. Journal of Transport Geography, 2016, 54: 248-256.

 Bayesian spatial-temporal model for the main and interaction effects of roadway and weather characteristics on freeway crash incidence. Accident Analysis and Prevention, 2019, 132, 105249.

RESPONSE: Thank you for the suggestion. The suggested references were added in the materials and methods section, line 212:

“For each application and data set observed, it is necessary to choose the most appropriate technique among the many available. In this case, different performance metrics can be used as selection criteria. Zeng and Wen used Mean Absolute Deviance (MAD) and Mean Squared Prediction Error (MSPE) [25,46], and, in an epidemiological context, Chowell used the Root Mean Squared Error (RMSE), Mean Squared Error (MAE), and Mean Absolut Percent Error (MAPE) [15].”

3 - In the ANN modeling section, a figure showing how the output and inputs connected would present the model more explicitly. 

RESPONSE: We have added the figure (Fig. 3) as suggested by the reviewer in the ANN modeling subsection.

4 - The Broyden-Fletcher-Goldfarb-Shanno iterative algorithm used for ANN training is rare in the previous studies. A detailed description of this algorithm is anticipated. 

RESPONSE: We have added a brief explanation of the BFGS algorithm and quote a reference that shows the detailed operation of the algorithm in the ANN modeling subsection, at line 178, as follow:

“The BFGS memoryless quasi-Newton was successfully used for minimizing errors on artificial neural networks. The Quasi-Newton method is a method that is used when the calculation of the Hessian matrix is difficult or time-consuming. This method has a rapid convergence when compared with the method of gradient descent [44].”

5 - It would be interesting and further demonstrate the advantages of ANNs to compare their performance with that of time series models that are used in the previous studies on forecasting COVID-19 cases.

RESPONSE: 

Thank you for the suggestion. Unfortunately, we did not found studies with similar methodology and proposal to compare. But we have added the information regards the work of Petropoulos F & Makridakis S. [54] that performs an analysis of different scenarios but with different day intervals and without goodness-of-fit analysis, which prevents a proper comparison.

---

## [Decision Letter · Decision Letter 1]

22 Feb 2021

Artificial neural networks for short-term forecasting of cases, deaths, and hospital beds occupancy in the COVID-19 pandemic at the Brazilian Amazon

PONE-D-20-38808R1

Dear Dr. Braga,

We’re pleased to inform you that your manuscript has been judged scientifically suitable for publication and will be formally accepted for publication once it meets all outstanding technical requirements.

Kind regards,

Qiang Zeng, Ph.D.

Academic Editor

PLOS ONE

Additional Editor Comments (optional):

Reviewers' comments:

Reviewer's Responses to Questions

**Comments to the Author**

1. If the authors have adequately addressed your comments raised in a previous round of review and you feel that this manuscript is now acceptable for publication, you may indicate that here to bypass the “Comments to the Author” section, enter your conflict of interest statement in the “Confidential to Editor” section, and submit your "Accept" recommendation.

Reviewer #1: All comments have been addressed

Reviewer #2: (No Response)

2. Is the manuscript technically sound, and do the data support the conclusions?

Reviewer #1: (No Response)

Reviewer #2: (No Response)

3. Has the statistical analysis been performed appropriately and rigorously? 

Reviewer #1: (No Response)

Reviewer #2: (No Response)

4. Have the authors made all data underlying the findings in their manuscript fully available?

Reviewer #1: (No Response)

Reviewer #2: (No Response)

5. Is the manuscript presented in an intelligible fashion and written in standard English?

Reviewer #1: (No Response)

Reviewer #2: (No Response)

6. Review Comments to the Author

Reviewer #1: (No Response)

Reviewer #2: (No Response)

7. PLOS authors have the option to publish the peer review history of their article (what does this mean?). If published, this will include your full peer review and any attached files.

Reviewer #1: No

Reviewer #2: No

---

## [Editor Report · Acceptance letter]

2 Mar 2021

PONE-D-20-38808R1 

Artificial neural networks for short-term forecasting of cases, deaths, and hospital beds occupancy in the COVID-19 pandemic at the Brazilian Amazon 

Dear Dr. Braga:

I'm pleased to inform you that your manuscript has been deemed suitable for publication in PLOS ONE. Congratulations! Your manuscript is now with our production department. 

Kind regards, 

on behalf of

Dr. Qiang Zeng 

Academic Editor

PLOS ONE